



# A simple, efficient, mass conservative approach to solving Richards' Equation (openRE, v1.0)

Andrew M. Ireson[1,2], Raymond J. Spiteri[3], Martyn P. Clark[4], Simon A. Mathias[5]

[1]Global Institute for Water Security, University of Saskatchewan, Saskatoon, Saskatchewan, Canada
[2]School for Environment and Sustainability, University of Saskatchewan, Saskatoon, Saskatchewan, Canada
[3]Department of Computer Science, University of Saskatchewan, Saskatoon, Saskatchewan, Canada
[4]Department of Geography and Planning, University of Saskatchewan, Saskatoon, Saskatchewan, Canada
[5]Department of Engineering, Durham University, Durham, UK

*Correspondence to*: Andrew M. Ireson (andrew.ireson@usask.ca)

**Abstract.** We show that a simple numerical solution procedure – namely the method of lines combined with an off-the-shelf ODE solver – can provide efficient, mass conservative solutions to the pressure-head form of Richards' equation. We present a novel method to quantify the boundary fluxes that reduces water balance errors without negative impacts on model runtimes. We compare this solution with alternatives, including the classic Modified Picard Iteration method of Celia et al. (1990) and the Hydrus 1D model (Šimůnek et al., 2005, 2016). We reproduce a set of benchmark solutions with all models. We find that Celia's solution has the best water balance, but it can incur significant truncation errors in the simulated boundary fluxes, depending on the time steps used. Our solution has comparable run-times to Hydrus and better water balance performance (though both models have excellent water balance closure for all the problems we considered). Our solution can be implemented in an interpreted language, such as MATLAB or Python, making use of off-the-shelf ODE solvers. We investigated alternative scipy ODE solvers in Python and make practical recommendations about the best way to implement them for Richards' equation. There are two advantages of our approach: i) the code is short and simple, making it ideal for teaching purposes; and ii) the method can be easily extended to represent alternative properties (e.g., novel ways to parameterize the $K(\psi)$ relationship) and processes (e.g., it is straightforward to couple heat or solute transport), making it ideal for testing alternative hypotheses.

## 1. Introduction

Richards' equation (RE) describes the movement of water in variably saturated porous media. Almost any practical application of RE requires a numerical solution, yet RE remains challenging to solve reliably and accurately for a given set of boundary conditions and soil hydraulic properties (Farthing and Ogden, 2017). RE has been extensively reviewed elsewhere (e.g., Vereecken et al., 2016; Farthing and Ogden, 2017, and references therein). RE has practical limitations in representing the flow processes in real soils containing macropores, especially with modelling infiltration and rapid percolation processes (Beven and Germann, 2013). The strength of RE is its ability to represent matrix drainage and capillary flows, which control evapotranspiration processes, and its ability to be coupled to heat and solute transport models.



For this reason, RE remains a common approach to simulate soil moisture in many terrestrial system models (e.g., vadose zone models, ecohydrology models, and land-surface models; Vereecken et al., 2016, Clark et al., 2021).

Our objective in this manuscript is to *i)* present a simple and practical approach to solve RE that is efficient, mass conservative, and uses open-source software (written in Python) that is readily available to everyone, including as a teaching
tool; and *ii)* present an improved mass balance calculation procedure for use with ordinary differential equation (ODE) solvers that apply adaptive-time stepping schemes (ATS). We investigate how to maximize the efficiency and accuracy of ODE solvers and provide guidance on the subtle challenges that arise in evaluating the boundary fluxes and the water balance. For our purposes, we define the following five criteria for success for an RE solver: *i)* successfully reproduce benchmark solutions for $\psi(t,z)$ and/or $\theta(t,z)$; *ii)* be mass conservative, with errors that are negligible based on the specific
application; *iii)* truncation errors in the simulated boundary fluxes should be tested and be negligible for the given time-step; *iv)* the solver should be computationally efficient; and *v)* all else being equal on criteria *i-iv*, the simplest code should be preferred. Simple code should be the most human readable/editable code, which means it should be clean, concise, modular, and free from redundancies.

The remainder of this paper is organized as follows. In section 2, we describe a simple yet powerful approach to solve RE numerically using ODE solvers that we implement and test in the Python and MATLAB programming languages. In section 2, we also discuss the complexities of closing the water balance in RE and present a novel solution that can be applied with any ODE solver: openRE. In section 3, we benchmark the performance of our proposed solution against existing numerical models and solutions, including Hydrus 1D, and the solution method proposed by Celia at al. (1990). In section 4, we
summarize our recommendations.

## 2. Solving Richards' Equation

RE is derived from the mass continuity equation applied to a control volume of soil, $\Delta x \Delta y \Delta z$ (L³), and for vertical flow
passing through the area $\Delta x \Delta y$ (L²), we have

$$\frac{\partial m}{\partial t} = -\frac{\partial (q\rho)}{\partial z} \qquad (1)$$

where $m$ (M L⁻³) is the mass of water per control volume of soil, $\rho$ (M L⁻³) is the density of water, $q$ (L T⁻¹) is the flux of water, $t$ (T) is time and $z$ (L) is depth below some fixed datum. Assuming that density is constant, we may write


$$\frac{\partial \theta}{\partial t} = -\frac{\partial q}{\partial z} \tag{2}$$

where $\theta$ ($L^3$ $L^{-3}$) is the volumetric water content, defined as the volume of water per control volume of soil. The vertical flux is given by Darcy's law,

$$q_z = -K(\psi)\frac{dh}{dz} = -K(\psi)\left(\frac{d\psi}{dz} - 1\right) \tag{3}$$

where $h$ (L) and $\psi$ (L) are the hydraulic head and matric potential head, respectively, and $K(\psi)$ (L $T^{-1}$) is the hydraulic conductivity. Combining Eqs. (2) and (3) we have

$$\frac{\partial \theta}{\partial t} = \frac{\partial}{\partial z}\left(K(\psi)\left(\frac{\partial \psi}{\partial z} - 1\right)\right) \tag{4}$$

which is the mixed form of RE (Celia et al., 1990). If we let $C(\psi) = d\theta/d\psi$ ($L^{-1}$), we can write

$$C(\psi)\frac{\partial \psi}{\partial t} = \frac{\partial}{\partial z}\left(K(\psi)\left(\frac{\partial \psi}{\partial z} - 1\right)\right) \tag{5}$$

which is the $\psi$ based form of RE (Celia et al., 1990). We can also express this as the $\theta$ form of RE, given by

$$\frac{\partial \theta}{\partial t} = \frac{\partial}{\partial z}\left(\frac{K(\theta)}{C(\theta)}\frac{\partial \theta}{\partial z} - K(\theta)\right) \tag{6}$$

where the constitutive relationships $C(\theta)$ and $K(\theta)$ are expressed as functions of $\theta$ (Celia et al., 1990). Changes in storage, $d\theta$, in Eqs. (4), (5), and (6) are associated with the filling and draining of soil pores. Because $\lim_{\psi \to 0^-} d\theta/d\psi = 0$, an elastic storage term is needed to solve RE in saturated, or close to saturated, conditions. This term represents the compression of the pore-water and the expansion of the pore-space as a function of increasing pore water pressure (though the latter is orders of magnitude larger than the former). The $\psi$ form of RE with elastic storage is written

$$\left(S_S\frac{\theta(\psi)}{\theta_S} + \frac{\partial \theta}{\partial \psi}\right)\frac{\partial \psi}{\partial t} = \frac{\partial}{\partial z}\left(K(\psi)\left(\frac{\partial \psi}{\partial z} - 1\right)\right) \tag{7}$$





where $S_S$ (L$^{-1}$) is the specific storage coefficient, and $\theta_S$ is the saturated water content, equal to the porosity (Kavetski et al., 2001). If $S_S$ is large it may have a non-negligible impact on the water balance, which needs to be accounted for (e.g., as in

Clark et al., 2021, Eq. 80). $S_S$ is often treated as a numerical smoothing factor for RE when conditions are saturated or close to saturation and its impact on the water balance is neglected (e.g., as in Tocci et al., 1997; Ireson et al., 2009). For convenience, hereafter we will define $C(\psi) = S_S \frac{\theta(\psi)}{\theta_S} + \frac{\partial \theta}{\partial \psi}$, such that elastic storage can be ignored by setting $S_S = 0$.

The $\psi$ form or mixed form of RE with elastic storage included can be shown to work for saturated conditions (Miller et al.,

1998) and so can be considered a general governing equation for flow in soils, aquitards, and confined and unconfined aquifers. The $\psi$ form of RE is widely reported not to be mass conservative (Milly, 1984, 1985, Celia et al., 1990, Farthing and Ogden, 2017), though it has been shown that mass conservative solutions are possible (Rathfelder and Abriola, 1994, Tocci et al., 1997). This issue will be explored in detail in this paper.

We consider a finite difference numerical solution for the $\psi$ form of RE that applies the method of lines to reduce the PDE in Eq. (7) to a system of ODEs of the form

$$\frac{d\boldsymbol{\psi}}{dt} = f(\boldsymbol{\psi}, t, \boldsymbol{z}) \qquad (9)$$

where $\boldsymbol{\psi}$ and $\boldsymbol{z}$ represent vectors containing discrete values of $\psi$ and $z$. A similar approach was taken by Tocci et al. (1997)

and Felder and Ogden (2017). There are three methods that can be applied to integrate Eq. (9) with respect to time:

- **One-step methods** with no iteration. Either forward stepping explicit methods or backward stepping implicit methods, where a single evaluation of Eq. (9) is performed for each time step. First order (Euler) or higher order methods in time are possible, but higher order methods introduce complexities when dealing with the water balance. Explicit methods are a simple starting point for new researchers to understand the problem but are not useful in

practice because they require very small time-steps. Semi-implicit one-step methods may employ the Thomas Algorithm, or similar, to solve the tri-diagonal matrix problem, but they do not iterate to improve the convergence of the problem, which is problematic due to the non-linear dependence of $K$ and $C$ on $\psi$ (Celia et al., 1990, reproduced below).

- **Iterative methods** that take one-step . Iterative backward stepping implicit methods are very widely used (Celia et

al., 1990, Rathfelder and Abriola, 2001). The iterations allow the solution to find the correct values of $K$ and $C$ at the next time step, as is required in a fully implicit solution. These methods can be applied to either the $\psi$ form or mixed form of RE. For the mixed form of RE, the convergence criterion can be based directly on water balance closure, as in Celia's Modified Picard Iteration method. For the $\psi$ form of RE, the convergence criterion is based on


$\psi$ values, and the solution is subject to larger water balance errors (Celia et al., 1990, Rathfelder and Abriola, 1994,
Felder and Ogden, 2017).

- **Adaptive time stepping (ATS) methods**. An implicit or explicit method is used to find $\psi_{t+\Delta t}$ over some *calculation* timestep; the truncation error in the solution is assessed, for example, by comparing two different-order solutions, as in Kavetski et al. (2001); and depending on the size of the error, the time step is either reduced to improve the accuracy or increased to improve the efficiency. The solution marches forward until reaching what we

term the *reporting* timestep, where the state variables are output. The advantage of this approach is that the states and fluxes calculated at the intermediate *calculation* steps contain useful information which can be exploited in the output, as we demonstrate in this paper. Kavetski et al. (2001, 2002a, 2002b) developed ATS solvers designed specifically to solve the different forms of RE, while other workers have applied "black-box" (Kavetski, et al., 2001) ODE or DAE (Differential Algebraic Equation) solvers to RE (Tocci et al., 1997; Ireson et al., 2009; Ireson

and Butler, 2013; Mathias et al., 2015, Clark et al., 2021). A wide range of ODE solvers are available in many different programming languages. Solutions are simple to implement and, as we show in this study, can outperform other methods in terms of accuracy and efficiency.

In this study, we implement each of the three possible methods for solving RE in scripts that are available from Ireson (2022,
https://github.com/amireson/openRE). All of the models in this study were coded in Python (version 3.8.11). We make use of the following libraries: *numpy* (version 1.20.3); *matplotlib* (version 3.4.2); *scipy* (version 1.7.1), which contains various ODE solvers, described below; and *numba* (version 0.53.1), which is a just-in-time compiler that is optional but speeds up the model runs considerably. We organize and run the models using makefiles (Jackson, 2016).

We also provide a MATLAB version of our recommended solution (implemented in MATLAB R2017b installed on a mac). The MATLAB implementation will not be described further, but compared with the optimal Python solution, the MATLAB model is somewhat simpler and achieves an equivalent performance in terms of simulated states and fluxes. We do not recommend Python over MATLAB (or vice versa) – both platforms work well, and the choice will come down to numerous factors, including, e.g., the availability of, and user familiarity with, either package or the need to use non-proprietary

software to satisfy open science requirements.

### 2. 1 One-step and iterative solutions

For the one-step and iterative methods, we have coded up the numerical solutions from Celia et al. (1990). This provides a
typical one-step method (first-order backward Euler implicit solution), and an excellent iterative method in their improved Modified Picard Iteration Method (MPM) solution. Fully reproducible details of these models were provided by Celia et al. (1990) and so will not be repeated here. These models were implemented in Python, and the code is provided at Ireson





(2022, https://github.com/amireson/openRE). The one-step model was coded up in 67 lines of code (note, blank lines and comment lines are not counted in the number of lines of code), with an additional 52 lines of code to configure the problem

(define the grid, hydraulic properties, etc.). The Picard Iteration method was coded up in 79 lines of code and required the same 52 lines of code to configure the problem. The Modified Picard Iteration was implemented using the numba just-in-time compiler (Appendix A.5) and was coded up in 90 lines of code, with an additional 59 lines of code to configure the problem. All solutions make use of the Thomas Algorithm to solve the tridiagonal linear system arising from the implicit method.


### *2.2 Adaptive time-stepping solution*

One major benefit of using a standard ODE solver to integrate RE is that the code is concise and easy to read and understand. Our objective is to write a function that will evaluate the right-hand side of Eq. (9) and feed this to the ODE solver. Within

this function the problem can be treated as instantaneous in time so that we only need to consider the spatial differences in our finite difference solution scheme. We will use a block-centered grid in space; i.e., state variables are stored at nodes located at the center of grid cells, while fluxes are defined at the block boundaries, as shown schematically in Figure 1. For simplicity here, we consider a uniform grid (constant $\Delta z$), but it is straightforward to adapt these solutions to non-uniform grids. We introduce two spatial indices (Figure 1): $i$ represents the nodal values, and $j$ represents the block boundaries, both

of which have an initial value of zero (since Python uses zero based indexing). Hence $j = i + 1/2$. We will start with the $\psi$ form of RE, for which the governing equation is given in the form

$$\left.\frac{\partial \psi}{\partial t}\right|_i = -\frac{1}{C(\psi_i)} \left.\frac{\partial q}{\partial z}\right|_i \tag{10}$$

and

$$\left.\frac{\partial q}{\partial z}\right|_i = \frac{q_{i+\frac{1}{2}} - q_{i-\frac{1}{2}}}{\Delta z} = \frac{q_{j+1} - q_j}{\Delta z} \tag{11}$$





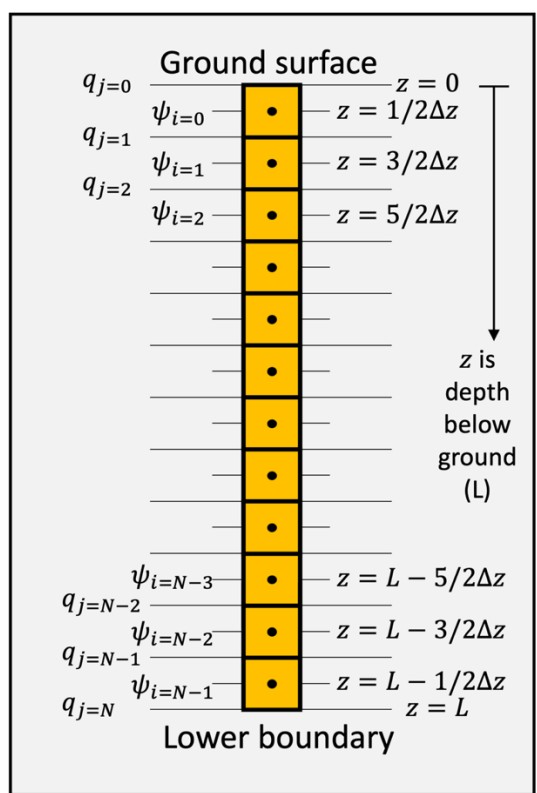

Figure 1. Schematic representation of the block centered finite difference grid, for a soil column of depth $L$, showing the zero-based Python indices of the states ($\psi_i$) and fluxes ($q_j$) on the left and depths of the nodes on the right, assuming a regular grid. There are $N$ states and $N + 1$ fluxes. The upper and lower boundary conditions are $q_{j=0}$ and $q_{j_N}$ respectively.


Here, the fluxes are given by

$$
\begin{aligned}
q_{i+\frac{1}{2}} = q_{j+1} &= -\frac{K(\psi_{i+1}) + K(\psi_i)}{2}\left(\frac{\psi_{i+1} - \psi_i}{\Delta z} - 1\right) \\
q_{i-\frac{1}{2}} = q_j &= -\frac{K(\psi_i) + K(\psi_{i-1})}{2}\left(\frac{\psi_i - \psi_{i-1}}{\Delta z} - 1\right)
\end{aligned}
\tag{12}
$$

In Eq (12), we are using the arithmetic mean of $K$ at the nodal points to estimate $K$ at the block boundaries, but other options
are possible (see, e.g., Bear and Cheng, 2010, p. 535). It is possible to combine Eqs. (10)-(12), but keeping them separate keeps the code modular and simple.

There are three commonly used boundary conditions for this problem, namely: i) a specified flux, $q_T$ (L T$^{-1}$) (type II boundary) is often used at the upper boundary to represent infiltration, where






$$q_{j=0} = q_T \tag{13}$$

ii) a free drainage boundary if often used at the lower boundary, where

$$q_{j=N} = K(\psi_{i=N-1}) \tag{14}$$

iii) a fixed $\psi$ (type I) boundary maybe used at the upper boundary, $\psi_T$ (L), typically to indicate a ponding depth, or at the lower boundary, $\psi_B$ (L), typically to represent a fixed water table, where

$$q_{j=0} = -\frac{K(\psi_T) + K(\psi_{i=0})}{2} \left( \frac{\psi_T - \psi_{i=0}}{\Delta z/2} - 1 \right)$$
$$q_{j=N} = -\frac{K(\psi_{i=N-1}) + K(\psi_B)}{2} \left( \frac{\psi_{i=N-1} - \psi_B}{\Delta z/2} - 1 \right) \tag{15}$$

Box 1 provides Python-based pseudo code that implements this solution (Eqs. 10 – 14) in a function for $d\psi/dt$ with a type II
boundary at the upper boundary and a free drainage boundary at the lower boundary and contains just 7 lines of code. This function can be called by the ODE solver.





```python
# Function inputs: psi,pars,dz,n,qT,psi_old
    # psi is an array containing the dependent variable
    # pars is a dictionary containing the soil parameters
    # dz is the space step
    # n is the number of spatial nodes
    # qT is the upper boundary flux
    # psi_old is psi at the previous time step (needed for
    # the mixed-form of RE)

# Function outputs: dpsidt
    # dpsidt is an array containing the derivative dpsi/dt

# Python based pseudo code:
    # Kfun function that returns the hydraulic conductivity
    # Cfun function that returns the specific storage

    # Calculate K and C at each node
    K=Kfun(psi,pars)
    C=Cfun(psi,pars,psi_old)

    # Initialize the flux array for grid boundaries:
    q=zeros[n+1]
    # Upper boundary condition:
    q[0]=qT
    # Fluxes for internal nodes:
    q[1:-1]=-(K[1:]+K[:-1])/2*((psi[1:]-psi[:-1])/dz-1)
    # Lower boundary condition:
    q[-1]=K[-1]
    # Continuity equation:
    dpsidt=-(1/C)*(q[1:]-q[:-1])/dz
```

Box 1. Python-based pseudo code implementation of RE with a block centered finite difference approach in a function to be called by an ODE solver. Arrays are zero-indexed; q[0] and q[-1] refer to the first and last item in the q array, respectively, and K[1:] and K[:-1] refer to a slice of the array K from the 2$^{nd}$ to the last node, and from the 1$^{st}$ to the 2$^{nd}$ to last node, respectively.







### 2.3 Mass balance closure

When RE is solved over some time interval, $t = t_0 \rightarrow t_M$ (where $t_M - t_0$ is typically multiple years in practical application)
for a soil profile $0 \leq z \leq L$, the cumulative inflow minus outflow should equal the change in storage in the profile over the
same interval; i.e., $\epsilon_B$ (mm), the bias error, defined in Eq. 16, should be zero.

$$\epsilon_B = \int_{t=t_0}^{t_M} \big(q(t,0) - q(t,z_N)\big)dt - \int_{z=0}^{L} \big(\theta(t_M,z) - \theta(t_0,z)\big)dz \tag{16}$$

The bias error can be treated as a mass balance performance metric for the model, but this metric may underestimate the true
errors in the water balance that occur within the time period simulated, which may cancel out over the entire run. The metric
used by Celia et al., (1990), Rathfelder and Abriola (1994), and Tocci et al. (1997) has the same problem. A more rigorous
mass balance performance metric is the root mean squared error of the daily (or some other time increment) cumulative net
flux minus the change in storage, $\epsilon_R$ (mm), where

$$\epsilon_R = \sqrt{\sum_{j=1}^{M} \left( \int_{t=t_j}^{t_{j+1}} \big(q(t,0) - q(t,z_N)\big)dt - \int_{z=0}^{L} \big(\theta(t_{j+1},z) - \theta(t_j,z)\big)dz \right)^2 / M} \tag{17}$$

where $j$ is an index in time, and $M$ is the number of timesteps considered. Reporting both metrics is informative – a high $\epsilon_R$
with a low $\epsilon_B$ indicates that daily errors are occurring but cancelling one another out; a high $\epsilon_B$ with a low $\epsilon_R$ indicates that
small daily errors are systematic in one direction and hence accumulate to give a high bias.

The fluxes in the mass balance calculation depend on the boundary conditions. For type I (specified $\psi$) and free drainage
type boundary conditions, the boundary flux depends on the simulated $\psi$ value at the node closest to the boundary. Over a
time increment $t_j$ to $t_{j+1}$, $\psi$ values will change continuously and hence so will the boundary flux. Due to the non-linearity of
the $K(\psi)$ relationship, the flux will change in a non-linear manner. The cumulative flux over the time increment, $Q_{j \rightarrow j+1}$
(mm), is given by

$$Q_{j \rightarrow j+1} = \int_{t=t_j}^{t_{j+1}} q(t)dt \tag{18}$$





$Q_{j \rightarrow j+1}$ is estimated from discrete values of $q$ (which in turn are approximated from discrete values of $\psi$, using either a forward difference approximation, where $Q_{j \rightarrow j+1} \approx q_j \Delta t$, a backward difference approximation (as in Celia et al., 1990), where $Q_{j \rightarrow j+1} \approx q_{j+1} \Delta t$, or a central difference approximation (as in trapezoidal integration), where $Q_{j \rightarrow j+1} \approx (q_j +$

$q_{j+1}) \Delta t / 2$. These discrete approximations for $Q_{j \rightarrow j+1}$ can be poor if the time step is large – or more precisely, if the changes in $q$ over a time step are large and non-linear.

This leads to an important limitation with Celia's mixed form solution to RE (and other equivalent iterative solutions). This solution has excellent mass balance closure, but because it uses a one-step iterative solution procedure with a backward

difference approximation for $Q_{j \rightarrow j+1}$, the simulated boundary fluxes can be shown to be sensitive to the time step (see Section 3.2). Hence, even though for larger $\Delta t$ the water balance is still perfectly closed, the actual terms within the water balance have changed, so there is less inflow and less change in storage.

ATS solvers can provide a practical solution to solving RE with good mass balance performance and boundary fluxes that do

not depend on the user specified timestep. The basic idea behind ATS solvers is that when there are large changes in $\psi$ in the model, small steps can be taken to capture the shape of $C(\psi)$ and $Q_{j \rightarrow j+1}$ and minimize the integration errors over a timestep. When the changes in $\psi$ are small, larger steps can be taken to maximize efficiency. We will refer to these adaptive time steps as *calculation* steps. The user specifies the time steps at which they wish the results to be saved, which we will refer to as the *reporting* time steps. In typical practical applications, the reporting step would be the resolution of the driving

data (e.g., hourly or daily). The solver may take many calculation steps of varying lengths between the reporting steps, saving the outcomes internally each time that the error tolerance is satisfied, and a successful step is taken.

To accurately calculate the boundary fluxes, it is necessary to use the $\psi$ information from the calculation time steps because $\psi$ may have evolved non-linearly over the reporting time step. However, this is not trivial. Two possible ways to do this

(which are equivalent to one another) are to *i)* enable dense output from the ODE solver (if this feature is supported by the ODE solver); or *ii)* force the ODE solver to make the reporting steps equal to the calculation steps. However, both of these approaches are more computationally demanding in terms of memory and runtime – a significant disadvantage. We propose here a third method for calculating the boundary fluxes that can be used with any ODE solver. At an instance in time, the cumulative boundary flux, $Q$ is related to the instantaneous flux, $q$ by


$$\frac{dQ}{dt} = q \qquad\qquad (19)$$





We can therefore use the ODE solver to integrate this expression and solve for $Q$. To do this, we add to the system of ODEs defined by Eq. 9 two new ODE expressions that represent the cumulative boundary fluxes. The dependent variable vector that is sent to the ODE solver now is $\boldsymbol{F}$, defined


$$\boldsymbol{F} = \begin{bmatrix} Q_{T:t_0 \to t} \\ \psi_0 \\ \psi_1 \\ \vdots \\ \psi_{N-2} \\ \psi_{N-1} \\ Q_{B:t_0 \to t} \end{bmatrix} \tag{20}$$

where $Q_{T:t_0 \to t}$ and $Q_{B:t_0 \to t}$ are the cumulative boundary fluxes at the current time, $t$, since the start of the simulation, $t_0$. The ODE solver will integrate the equation

$$\frac{d\boldsymbol{F}}{dt} = f(\boldsymbol{F}, t, \boldsymbol{z}) \tag{21}$$

to solve for $\boldsymbol{F}$. The function that is called by the ODE solver will evaluate the vector

$$\frac{d\boldsymbol{F}}{dt} = \begin{bmatrix} \dfrac{dQ_{T:t_0 \to t_j}}{dt} \\ \dfrac{d\psi_0}{dt} \\ \dfrac{d\psi_1}{dt} \\ \vdots \\ \dfrac{d\psi_{N-2}}{dt} \\ \dfrac{d\psi_{N-1}}{dt} \\ \dfrac{dQ_{B:t_0 \to t_j}}{dt} \end{bmatrix} = \begin{bmatrix} q_T \\ -\dfrac{1}{C(\psi_0)} \left.\dfrac{\partial q}{\partial z}\right|_0 \\ -\dfrac{1}{C(\psi_1)} \left.\dfrac{\partial q}{\partial z}\right|_1 \\ \vdots \\ -\dfrac{1}{C(\psi_{N-2})} \left.\dfrac{\partial q}{\partial z}\right|_{N-2} \\ -\dfrac{1}{C(\psi_{N-1})} \left.\dfrac{\partial q}{\partial z}\right|_{N-1} \\ q_B \end{bmatrix} \tag{22}$$

Note that each term in Eq. (22) is expressed at a single instant in time, $t$, and subscripts $0, 1, \ldots, N-2, N-1$ refer to the indices of the finite difference discretization points, and we use zero-based indexing to be consistent with the Python

language and Figure 1. After solving for $\boldsymbol{F}$, the first and last rows of $\boldsymbol{F}$ correspond to the cumulative boundary fluxes, which at time $t$ are $Q_{T:t_0 \to t}$ and $Q_{B:t_0 \to t}$. The cumulative fluxes over each time step are obtained from


$$Q_{T:t_j \to t_{j+1}} = Q_{T:t_0 \to t_{j+1}} - Q_{T:t_0 \to t_j}$$
$$Q_{B:t_j \to t_{j+1}} = Q_{B:t_0 \to t_{j+1}} - Q_{B:t_0 \to t_j}$$

(23)

Solving systems of ODEs in this way is straightforward – requiring the user to pack and unpack the dependent variable
vector. Python based pseudo code showing how this can be implemented is given in Box 2. Note, here we multiply the fluxes
by 1000 within the solver (equivalent to converting the fluxes from m/d to mm/d), so that the magnitude of the fluxes is
comparable with the magnitude of the changes in $\psi$ (typically in m), which can improve the water balance estimate. Any
arbitrary scaling factor can be applied here, as long as the output fluxes are scaled back to the correct units. We shall refer to
this method for calculating the boundary fluxes as the *solver flux output method* (SFOM).




```
# ODE function inputs: tSpan,F,args
    # tSpan array with the start and end date for the integration
    # F array with the dependent variable
    # args list of other variables (see Box 1)

# ODE function outputs: dFdt
    # dFdt array with the time derivative of the dependent variable

# Python based pseudo code for the ODE function:
    # Unpack the dependent variable:
    QT=F[0]/1000.      # Cumulative flux at the upper boundary
    QB=F[-1]/1000.     # Cumulative flux at the lower boundary
    psi=F[1:-1].       # matric potential at each node

    # Insert here the contents of the function in Box 1, which
    # calculates dpsidt, q[0] and q[-1]

    # Pack up the dependent variable in a single column array:
    dFdt=hstack([q[0]*1000.,dpsidt,q[-1]*1000.])

# Pseudo code to call the ODE solver:
    # Initialize the 2D array (t,z) for the dependent variable
    F=zeros(nt,nz)

    # Pack up the initial condition:
    F[0,0]=0.    # Cumulative flux at the upper boundary
    F[0,1:-1]=psiI
    F[0,-1]=0.   # Cumulative flux at the lower boundary

    # Solve using a time loop:
    for j in 1 to M-1:
        F[j+1,:]=CallODE(ODEfunction,tSpan=[0,dt],F[j,:],args)

    # Unpack dependent variable:
    QT=hstack([0,F[1:,0]-F[:-1,0]])/1000.
    psi=F[:,1:-1]
    QB=hstack([0,F[1:,-1]-F[:-1,-1]])/1000.
```

Box 2. Python-based pseudo code implementation of an ODE solver solution. The ODE function returns the time derivative of the dependent variable array, given in Eq. 19. Note, F[:,1:-1] is a slice through the array F that takes all the items in the first (time) dimension and the 2nd to 2nd to last items in the second dimension.






## 2.4 Improving efficiency

Here, we provide details of two techniques that can be used to improve the computational runtime of the model. These methods have no impact on the accuracy of the solution, so are optional, but combined they can result in better than a factor
of 10 reduction in the runtime at the cost of only a few additional lines of code. In Appendix A, we investigate the impact of a range of different possible model decisions/assumptions on the accuracy, mass balance, efficiency, and simplicity of the model.

### 2.4.1 Defining the Jacobian Pattern

For the form of RE given by Eq. 10, the Jacobian matrix, $J$, is an $n \times n$ matrix, where the cell in each row, $i$, and column, $j$, is defined by the derivative

$$J_{i,j} = \frac{d}{d\psi_j}\left(\frac{d\psi}{dt}\bigg|_i\right) = \frac{d}{d\psi_j}\left(-\frac{1}{C(\psi_i)}\frac{\partial q}{\partial z}\bigg|_i\right) \tag{24}$$

Each entry in $J_{i,j}$ can be evaluated in a function that is passed to the ODE solver (assuming that the particular ODE solver being used has this functionality), in order to speed up the solution process. For the spatial discretization scheme described above, all the terms of the Jacobian are zero, except for where $j = i - 1, j = i$, and $j = i + 1$ (ignoring when $i - 1 < 1$ and $i + 1 > n$, which would be zero terms anyway, for any boundary condition). A simpler alternative to defining the full Jacobian matrix is to define the *Jacobian Pattern*. The Jacobian Pattern is a matrix of ones and zeros that defines the location
of the structurally non-zero elements of the Jacobian – that is, where the terms are not identically zero. For the spatially discretized RE as given in Eq. 10, the Jacobian Pattern is a simple tridiagonal matrix, with ones on the three main diagonals, and zeros everywhere else. To implement this requires an ODE solver capable of using the Jacobian Pattern (also referred to as the Jacobian sparsity matrix). The *scipy* ODE solvers *ode* and *solve_ivp* have the ability to define a banded Jacobian pattern: setting *uband* and *lband* arguments to 1 tells the ODE solver that the Jacobian is a tri-diagonal matrix. The *scipy*
ODE solver *solve_ivp* can also handle a general $n \times n$ Jacobian pattern, which is more adaptable for multi-dependent variable coupled problems (e.g., Goudarzi et al., 2016). The MATLAB ODE solvers can read the Jacobian sparsity pattern matrix from the JPattern argument. We report on the relative performance/complexity of each of these methods in Appendix A.4.






### *2.4.2 Just-in-time compilation*

In this paper, we are providing guidance for the use of interpreted programming languages (e.g., Python or MATLAB) to solve RE. Interpreted languages have a number of advantages over compiled languages (such as FORTRAN and C), including that: they are easier to learn, with excellent teaching resources widely and freely available; they tend to have higher level abstractions, so that the same task can be completed in fewer lines of code; and they are cross platform and typically easier to install on any computer. Interpreted languages are compiled on the fly, meaning every individual line of

code is compiled at run-time. By contrast, compiled languages are more efficient because the compilation and running of the code are separated, and you typically only have to compile the code once before it can be run many times. A nice compromise between the simplicity of interpreted languages and efficiency of compiled languages is to use a just-in-time compiler. In Python, the *numba* library is such a just-in-time compiler. Numba compiles all the lines of the Python code once at the start of the runtime, and then all subsequent calls to the code run much faster. We find that using *numba* in conjunction

with our preferred ODE solver solution described above, results in up to 10x faster code execution (see Appendix A.5). The drawback to using *numba* is that some re-factoring of the code may be necessary to make a script that previously ran without *numba* work using *numba* – in particular, there are complications around how variables are allocated into *numpy* arrays. We include code in Ireson (2022, https://github.com/amireson/openRE) that demonstrates how to successfully implement *numba*.


### 3. Benchmarking the model performance

In this section, we run our RE solver, openRE, on a number of benchmark problems, comparing the different solution procedures and assessing the performance of all solutions against the five success criteria identified in the introduction,

namely: *i)* accuracy of $\theta(t,z)$ and $\psi(t,z)$; *ii)* mass balance performance; *iii)* consistency boundary fluxes with $\Delta t$; *iv)* computational efficiency (i.e., runtime); and *v)* simplicity of the code.

### 3.1 Published model benchmarks

### *3.1.1 Celia's problem*

Celia's test case (Celia et al., 1990) is used to compare our ATS solution with the different solutions schemes previously proposed by Celia et al. (1990). The test problem uses a 40 cm deep vertical soil profile ($z_N = 40$) with a uniform 1 cm space step ($dz = 1$), a 360 second duration ($t_0 = 0$; $t_M = 360$) with a 1 second time step ($\Delta t = 1$), and the following initial and type I boundary conditions:






$$\psi(t = t_0, 0 \leq z \leq z_N) = -61.5 \text{ cm}$$
$$\psi(t_0 \leq t \leq t_M, z = 0) = -20.7 \text{ cm} \tag{25}$$
$$\psi(t_0 \leq t \leq t_M, z = z_N) = -61.5 \text{ cm}$$

The soil hydraulic properties are given by

$$\theta = \frac{\alpha(\theta_s - \theta_r)}{\alpha + |\psi|^\beta} + \theta_r$$
$$K = K_s \frac{A}{A + |\psi|^\gamma} \tag{26}$$

where the parameter values are $\alpha = 1.611 \times 10^6$, $\theta_s = 0.287$, $\theta_r = 0.075$, $\beta = 3.96$, $K_s = 0.00944$ cm/s, $A = 1.175 \times 10^6$, and $\gamma = 4.74$. Celia et al. (1990) presented three solution schemes: the "No iteration scheme" uses the $\psi$ form of RE and solves the problem with a single backward implicit step and no iteration (which we achieved using the Thomas Algorithm); the "Picard Iteration scheme" also solves the $\psi$ form of RE, but uses the Picard iteration method to improve the solution, with errors in $\psi$ used as a convergence criterion; finally the "Modified Picard Iteration Method" (MPM) uses the

mixed-form of RE and uses errors in $\theta$ as a convergence criterion. The MPM is mass conservative because the iteration ensures that the cumulative change in fluxes (the right-hand-side of RE) balance the changes in storage (the left-hand-side of the mixed form of RE). Celia's three solutions were reproduced in Python scripts (https://github.com/amireson/openRE) and compared with our ATS/SFOM solution. Results from all three solutions are shown in Figure 2. All methods are consistent for very small timesteps. The one-step method with no iteration performs poorly, with delayed breakthrough of the wetting

front when the time step is large. The solution is improved by using Picard iteration, but there are still some delays. The MPM has a much better performance but, as with all implicit Euler time-stepping schemes, is still subject to some numerical dispersion for larger timesteps (van Genuchten et al., 1978). The ATS solution reproduces the $\psi$ breakthrough curve but with no dispersion and no differences associated with the timestep. Note that the time steps for plotting the ATS solution only represent the *reporting* time step - the underlying *calculation* time steps – are likely much smaller (Section 2.2).




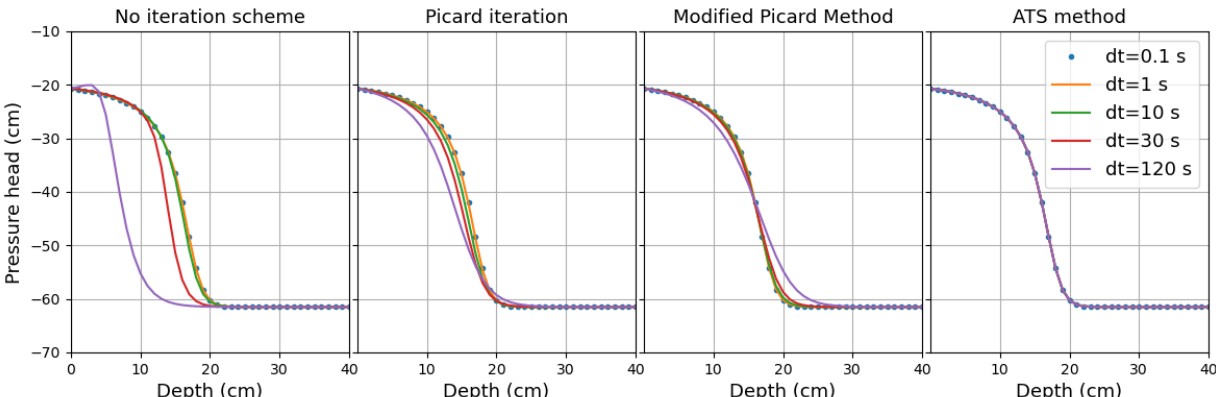

Figure 2. Reproduction of Celia's model benchmark: the $\psi$ breakthrough curve using a backward difference implicit Euler solution with no iteration scheme; backward difference implicit Euler with Picard iteration; Celia's modified Picard iteration Method (MPM); and the ODE solver with adaptive time stepping. The ODE solver produces consistent $\psi(t, z)$ results independently of the mass balance calculation procedure. The time steps reported are calculation time steps for Celia's solutions but are reporting timesteps for the ODE solver, which uses an adaptive time step for calculation steps.

In Figure 3, we show the cumulative inflow simulated by each of these models for Celia's benchmark problem, along with the mass balance bias error, for different reporting timesteps. The one-step solution with no iteration and the Picard Iteration solution both have poor mass balance performance unless the time step is very small – on this basis, we do not consider these solutions further. We see that the MPM method is perfectly mass conservative for any $\Delta t$ used in the model, as we should expect. However, we can see that the cumulative inflow is sensitive to $\Delta t$. Hence, even though for larger $\Delta t$ the water balance is still perfectly closed, the actual terms within the water balance have changed, so there is less inflow and less change in storage. This is perhaps an under-appreciated limitation of Celia's MPM solution, and solutions to the mixed form of RE generally – which is that mass balance is a necessary but insufficient criterion for model performance assessment, and truncation errors can still be present in the fluxes with perfect water balance closure.

In Figure 3, we also show the water balance performance of our ATS solution, using either reporting time step information for the water balance calculation or using calculation time step information (i.e., using the SFOM described in Section 2.3). Using reporting timestep information is the easiest and most intuitive approach to take – you numerically integrate (1) discrete $\theta$ values over depth to get storage and (2) discrete $q$ values during reporting timesteps to get $Q$ (e.g., using trapezoidal integration). However, this approach fails to capture non-linear changes in $q$ over a reporting timestep and results in large water balance errors and errors in the cumulative fluxes, as is clear in Figure 3. Using the SFOM, we see that the water balance is almost exactly closed, and the boundary fluxes are independent of the reporting timestep. It is also



important to note that the discrete $\psi(t, z)$ values simulated by both ATS solution procedures here are identical (see Figure 2) – the only difference is how the boundary fluxes are calculated.

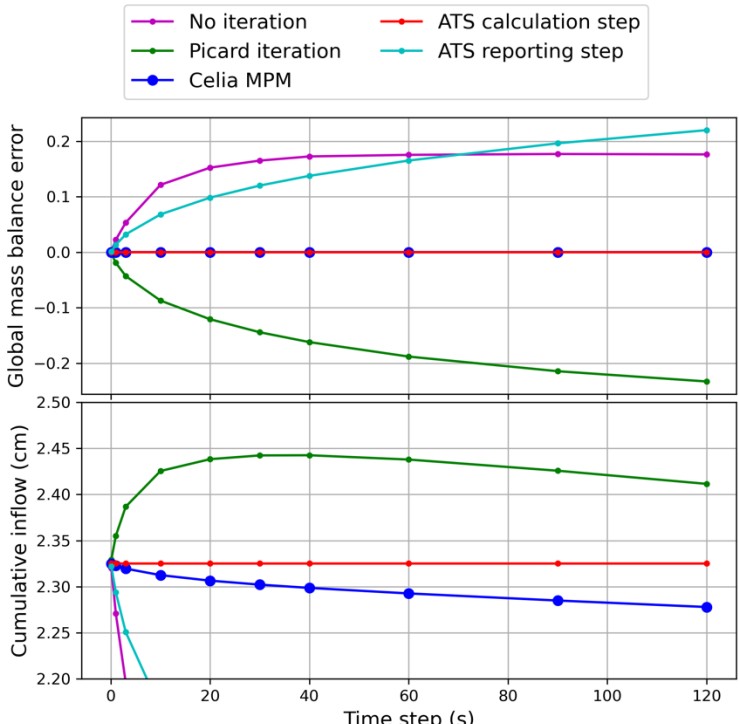

Figure 3. Mass balance of Celia's three models and our adaptive time-stepping model. The upper plot shows a reproduction

of Celia's mass balance calculation. The lower plot shows the cumulative inflow that is simulated.

### 3.1.2 Miller's saturated infiltration pulse problem

Miller et al. (1998) investigated solutions to RE that aimed to address numerical convergence problems associated with challenging boundary conditions, and they specifically looked at the problem of infiltration from a ponded upper boundary into a hydrostatic soil profile with a fixed water table at the lower boundary. This is a good benchmark because it requires the model to deal with perched saturated conditions over unsaturated conditions and involves highly non-linear changes in properties over short distances and time steps. The problem uses van Genuchten (1980) hydraulic properties, given by

$$S_e = (1 + (\alpha\psi)^n)^{-m} \tag{27}$$





$$\theta = \theta_r + (\theta_s - \theta_r)S_e \tag{28}$$

$$\frac{d\theta}{d\psi} = \frac{-\alpha m(\theta_s - \theta_r)}{1-m} S_e^{1/m} \left(1 - S_e^{1/m}\right)^m \tag{29}$$

$$K = K_s S_e^{1/2} \left(1 - \left(1 - S_e^{1/m}\right)^m\right) \tag{30}$$


where $S_e$ (-) is the effective saturation, $\alpha$ (L$^{-1}$), $n$ (-) and $m$ (-) are parameters that determine the shape of the $\theta(\psi)$ curve, $\theta_r$ (-) and $\theta_s$ (-) are the residual and saturated volumetric water contents and $K_s$ (L T$^{-1}$) is the saturated hydraulic conductivity. Miller's problem uses the parameters in Table 1 for three different soil types.

| Parameter | Sand | Loam | Clay Loam |
|:---:|:---:|:---:|:---:|
| $\theta_r$ | 0.093 | 0.078 | 0.095 |
| $\theta_s$ | 0.301 | 0.430 | 0.410 |
| $\alpha$ (1/m) | 5.470 | 3.600 | 1.900 |
| $n$ | 4.264 | 1.560 | 1.310 |
| $K_s$ (m/d) | 5.040 | 0.250 | 0.062 |
| $S_s$ (1/m) | $10^{-6}$ | $10^{-6}$ | $10^{-6}$ |

425                 Table 1. Soil hydraulic properties used the Miller et al. (1998) problem

A hydrostatic initial condition is used, with a fixed water table at depth of 10, 5, and 2 m below ground surface for sand, loam, and clay loam, respectively. At the upper boundary, 0.1 m of ponding is applied throughout the simulation run time of 0.18, 2.25, and 1.0 days for sand, loam, and clay loam, respectively. We simulated this problem with our ATS solution and with Celia's MPM model, for comparison purposes. Both models faced challenges with this problem. For Celia's MPM, we

had to use a small timestep to get the solver to produce accurate $\psi(z)$ profiles (Figure 4). For the ATS solutions using the default ODE solver settings, the models failed to propagate the wetting front into the soil correctly. It was necessary to increase the maximum number of calculation steps allowed per reporting time step (we increased this from the default 500 to 10,000) so that very small timesteps could be taken (Table 2). It was also necessary (for loam and clayloam) or beneficial

(for sand) to reduce the solver error tolerances – see values in Figure 4 and Table 2. The results from these simulations are shown in Figure 4 and are consistent with those reported in Figure 1 of Miller et al. (1998), showing that both models are able to successfully reproduce this benchmark. The runtimes and water balance for each solution are tabulated in Table 2. Celia's MPM has consistently better water balance performance, though we think the water balance errors in both models are acceptably low. The ATS solution is slower for sand, faster for loam, and about the same for clayloam. Note that the

runtimes and water balances of the ATS solution are sensitive to the reporting timestep $\Delta t$ and the solver settings nsteps,
atol, and rtol – an improved solution might be attainable by optimizing these settings. On the other hand, for Celia's MPM
solution we only needed to optimize $\Delta t$.

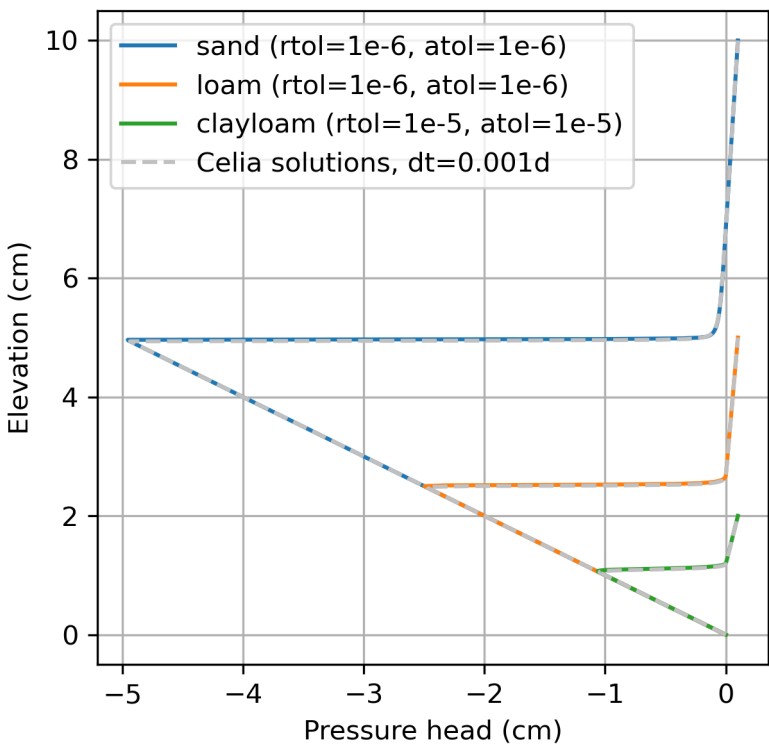

Figure 4. Reproduction of the Miller infiltration pulse result, using Celia's MPM model and our ATS SFOM solution. Both
445        models are satisfactorily consistent with the output reported by Miller et al. (1998), in their Figure 1.





| Solution | Soil | Solver settings | runtime (s) | MB bias (mm) | MB RMSE (mm) |
|---|---|---|---|---|---|
| Celia MPM | Sand | $dt = 0.001$ d | 4.4 | -4.2E-06 | 4.5E-06 |
| | Loam | $dt = 0.001$ d | 31.7 | -1.6E-03 | 5.3E-06 |
| | Clay loam | $dt = 0.001$ d | 32.6 | -8.0E-02 | 2.5E-04 |
| ATS SFOM | Sand | `atol/rtol` $= 10^{-6}$<br>`nsteps` $= 10,000$<br>$dt = 0.01$ | 11.7 | -1.5E-02 | 8.8E-04 |
| | Loam | `atol/rtol` $= 10^{-6}$<br>`nsteps` $= 10,000$<br>$dt = 0.01$ | 16.7 | -8.8E-02 | 2.0E-03 |
| | Clay loam | `atol/rtol` $= 10^{-5}$<br>`nsteps` $= 10,000$<br>$dt = 0.001$ | 30.8 | -3.2E-01 | 1.3E-03 |

Table 2. Runtime and water balance performance for the Celia MPM and ATS SFOM solutions, applied to the problem in Miller et al., (1988). We also show here the solver settings. For the ATS solutions, we had to reduce the relative and absolute tolerances (`atol/rtol`) of the integrator and increase the maximum number of calculations steps allowed within a reporting step (`nsteps`). Note, timesteps ($dt$) in ATS solutions are reporting timesteps.

### 3.1.3 Mathias' solution for horizontal infiltration

Mathias and Sander (2021) developed a pseudospectral similarity solution for horizontal infiltration (i.e., solving RE without gravity) that is fast and accurate. This solution assumes a semi-infinite horizontal soil column with a uniform initial condition and a type 1 boundary condition on the left boundary. The solution can resolve very large gradients in saturation and $\psi$ at the boundary, which propagate into the soil rapidly – and as such this is another challenging problem for a numerical RE model to reproduce. We solved this problem for three soil types, namely Hygiene sandstone, silt loam G.E.3, and Beit Netofa Clay, with properties from van Genuchten (1980), as listed in Table 3. $S_s$ was set to 0, consistent with Mathias' solution. To configure our model for horizontal flow, it is necessary to remove the gravity term from the flux calculations (i.e., Eq. 12). We solved this problem for a left-hand boundary effective saturation of 0.99 and an initial saturation of 0.01. The grid is configured such that the wetting pulse does not reach the right-hand boundary over the simulation runtime.





| Parameter | Sand | Loam | Clay Loam |
|---|---|---|---|
| $\theta_r$ | 0.153 | 0.131 | 0. |
| $\theta_s$ | 0.250 | 0.396 | 0.446 |
| $\alpha$ (1/cm) | 0.0079 | 0.00423 | 0.00152 |
| $n$ | 10.4 | 2.06 | 1.17 |
| $K_s$ (cm/d) | 108. | 4.96 | 0.082 |
| $S_s$ (1/m) | 0. | 0. | 0. |
| $dt$ (min) | 0.1 | 0.01 | 0.01 |
| $dx$ (cm) | 0.25 | 0.05 | 0.0025 |


Table 3. Soil hydraulic properties used the Miller et al. (1998) problem

We solved this problem with our ATS solution, with Celia's MPM solution, and with the pseudospectral similarity solution (Mathias and Sander, 2021, implemented in MATLAB). The results in Figure 5 shows that both the ATS solution and the Celia solution do an excellent job of reproducing this solution for $\theta(t, x)$ (where $x$, m, is horizontal distance). The runtimes

and water balance for each solution are tabulated in Table 4. Here we see that the ATS and Celia solutions have the same performance in terms of the water balance and the cumulative fluxes simulated. Runtimes vary between models: both are the same for sandstone, Celia's solution is faster for silt loam, while the ATS solution is faster for clay.





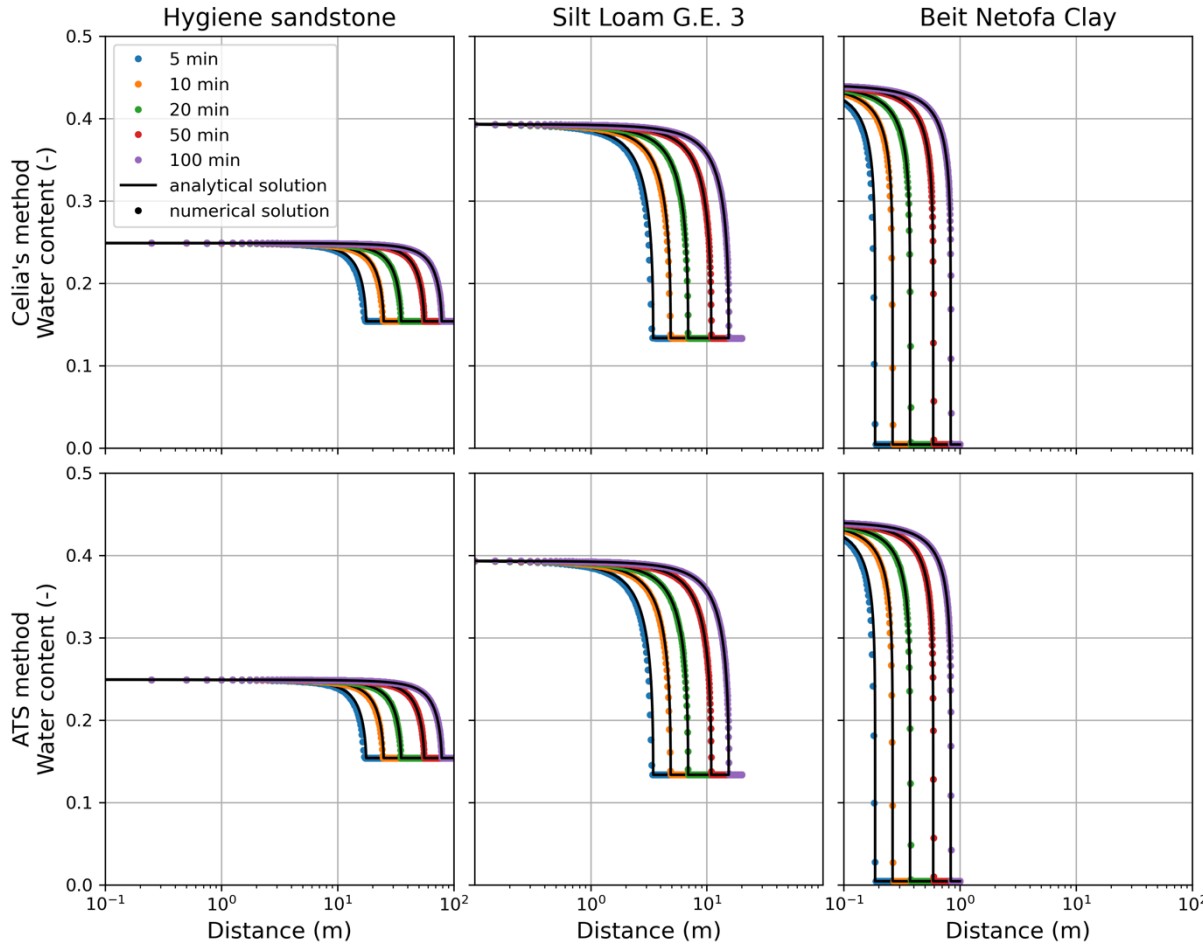


Figure 5. Comparison of our ATS solver flux output model for horizontal infiltration with the Mathias and Sander (2021) pseudospectral similarity solution (denoted analytical in the legend).




| Solution | Soil | runtime (s) | MB bias (mm) | MB RMSE (mm) | Cumulative infiltration (mm) |
|---|---|---|---|---|---|
| Celia MPM | Sandstone | 2.4 | 4.0E-03 | 6.5E-06 | 63.2 |
| | Silt loam | 12.8 | 2.2E-02 | 3.8E-06 | 34.2 |
| | Clay | 69.4 | 2.2E-03 | 4.1E-07 | 3.4 |
| ATS SFOM | Sandstone | 2.3 | -1.1E-03 | 4.8E-06 | 63.3 |
| | Silt loam | 18.8 | 1.9E-02 | 3.9E-06 | 34.2 |
| | Clay | 32.1 | 2.2E-03 | 3.9E-07 | 3.4 |

Table 4. Runtime and water balance performance for the Celia MPM and ATS SFOM solutions, applied to the problem described by Mathias and Sander (2021).

## 3.2 Comparison with Hydrus 1D


Hydrus 1D (Šimůnek et al., 2005, 2016) is a widely used one-dimensional RE solver. The calculations within Hydrus are undertaken using openly available FORTRAN source code, and the software runs through a (closed-source) graphical user interface on Microsoft Windows. The FORTRAN code can be compiled using gfortran on the MacOS operating system and run from the command line, which we did here, so that the runtime comparisons with our model are fair. Within HYDRUS,

the user interface provides somewhat limited control over the error tolerances. We were unable to modify any settings to improve the water balance, and so we present here model runs that use the default iteration criteria (maximum number of iterations = 100; water content tolerance = 0.001; pressure head tolerance = 10 mm; lower/upper optimal iteration range = 0.7/1.3; lower/upper timestep multiplication factor = 1.3/0.7; lower/upper limit of tension interval = $10^{-6}/10^3$ cm).

We configured Hydrus 1D, our ATS solution, and our implementation of Celia's MPM solution for a simple numerical experiment, where we simulate the infiltration of a ten-year timeseries of daily precipitation into a 1.5 m deep soil column, with a free drainage lower boundary condition. The minimum, mean, and maximum annual precipitation was 265, 484, and 680 mm/yr, and the maximum daily precipitation was 55 mm/d. We used Silt Loam G.E. 3 soil hydraulic properties from van Genuchten (1980), where $\theta_r = 0.131$, $\theta_s = 0.396$, $\alpha = 0.423$ m$^{-1}$, $K_s = 0.0496$ m/d, $n = 2.06$. We set $S_s = 10^{-6}$ m$^{-1}$

and used a uniform $\psi$ initial condition of -3.59 m. The results of the simulations with each model are given in Table 5,





showing the runtime and water balance performance, Figure 6, showing the detailed water balance performance, and Figure 7, showing the simulated storage and drainage.

In our ATS solution, we can trade-off between water balance error and the runtime by modifying the rtol argument for the
ODE solver. We found that the default rtol of $10^{-6}$ had the fastest runtime, but the water balance performance, whilst good enough for all practical purposes, was the worst overall (Table 5, Figure 6). Therefore, we reduced rtol to $10^{-7}$, which improved the water balance performance but increased the runtime. Even though the water balance errors reported here are all very small, it is still interesting to look closely at how these compare for the different models, as shown in Figure 6. The first thing to note is that the Celia MPM solution has water balance errors of essentially zero, which we expect, because this
solution enforces water balance closure. Celia's solution did have the longest runtime – approximately 40% slower than the other solutions. In the ATS solutions, on a daily basis, the water balance errors are much smaller than in Hydrus. However, in Hydrus, the water balance errors appear random, with a mean of zero, and hence when looking at the cumulative errors in Hydrus, there is no systematic accumulation in the errors. In the ATS solutions, in the lower right plot of Figure 6, we can see that the water balance errors are strongly correlated to the infiltration flux at the upper boundary – larger fluxes result in
larger errors. Hence for the ATS solution with rtol=$10^{-6}$, we see that the errors accumulate, and after 4 years, the cumulative errors in the ATS solution exceed those in Hydrus. For the ATS solution with rtol=$10^{-7}$, the errors do not accumulate monotonically, and the long-term cumulative errors tend to oscillate about zero. The water balance performance of the ATS solution with rtol=$10^{-7}$ is therefore better than the performance in Hydrus (Table 5, Figure 6), while the runtimes of these models are essentially the same (Hydrus is slightly faster, with a runtime of 2.21 vs 2.30 seconds, Table 5).


Looking at the simulated storage and discharge in Figure 7, the two ATS solutions are visually indistinguishable and are both broadly consistent with the Hydrus 1D model outputs. The Celia MPM solution has non-negligible differences with all other solutions. This is because the MPM solution applies an iterative solution procedure to solve the model at a daily timestep, and the boundary fluxes are therefore subject to errors, as discussed above. The solution scheme imposes mass balance on
the problem but does not track the truncation errors in the fluxes. The avoidance of this issue represents a significant advantage of adaptive time stepping solutions.

| Model | runtime (s) | MB bias (mm) | MB RMSE (mm) |
|---|---|---|---|
| ATS solution, rtol=$10^{-6}$ | 1.71 | -0.018 | 8.06E-05 |
| ATS solution, rtol=$10^{-7}$ | 2.30 | 0.0003 | 6.92E-05 |
| Celia MPM solution | 3.2 | 0.0 | 2.3E-10 |
| Hydrus | 2.21 | -0.0021 | 4.05E-03 |

Table 5. Comparison of model runtimes and mass balance performance.



Figure 6. Comparison of water balance performance from Hydrus 1D, the ATS solution, and our implementation of Celia's MPM solution. The water balance error is $q_T - q_B - \Delta S$, and we show the balance for each time step (top) and cumulative balances since the start of the simulation (bottom).

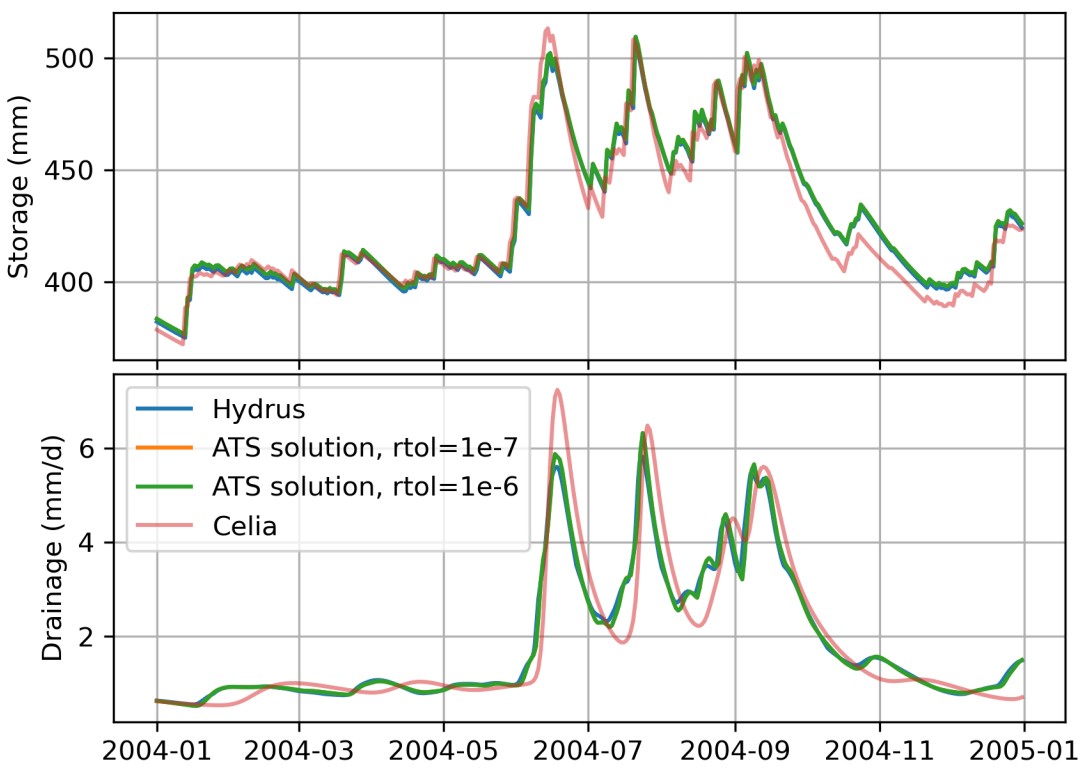


Figure 7. Simulated storage and discharge using Hydrus 1D, the ATS solution and our implementation of Celia's MPM
solution.

## 4. Summary and recommendations


We developed a simple adaptive time-stepping scheme (ATS) for RE using the interpreted language Python and making use of the *scipy* ODE solver *ode*. We also developed a new solver flux output method (SFOM) whereby cumulative boundary fluxes can be included within the dependent variable vector, allowing the determination of highly accurate integrated fluxes over designated time periods. The SFOM is particularly useful for providing reliable assessment of mass balance closure. In
principle, SFOM can be implemented in any ODE solver because it does not require any special output (such as dense output) to be available). Our model was coded up in python and released with the name openRE (Ireson, 2022). Our model performed well against our five success criteria, namely: *i)* we successfully reproduced benchmark solutions for $\psi(t,z)$ and $\theta(t,z)$ from Celia et al., (1990), Miller et al. (1998) and Mathias and Sander (2021); *ii)* we report negligibly low mass





balance errors; *iii)* we simulate boundary fluxes that are independent of the reporting time step (unlike Celia's solution, as

demonstrated in Figure 7); *iv)* we have low runtimes (as good as Hydrus 1D), and *v)* our code is very simple, concise (92 lines of code for the solver plus 68 lines of code for model configuration for the numerical experiment in Section 3.2), and easily adaptable to new problems. Our solution had the best balance of efficiency, accuracy, and simplicity as compared to alternative established solution procedures.

**Appendix A: Navigating pitfalls in ODE solver solutions**

There are several subtle decisions that must be made when solving RE using a generic ODE solver. Here, we test a number of alternative model configurations and report the impact of these decisions using the following metrics: for model accuracy (criteria *i*), we report the RMSE of $\psi$ at all grid points in $t$ and $z$ between the current model run and a reference model run;

for the mass balance (criteria *ii*), we report both the bias error (Eq. 16) and the more rigorous daily water balance RMSE (Eq. 17); for the model efficiency (criteria *iv*), we simply report the runtime, where all runs were undertaken on the same laptop computer. For these numerical experiments, we used the ten-year infiltration numerical experiment described in Section 3.2.

The best model configuration, against which all other model configurations are compared, was as follows: use the scipy

ODE solver "ode" with the method "BDF" (backward differentiation formulas, Brown et al., 1989); use our SFOM solution (Section 2.3/Appendix A.2); use the analytical expression for $C(\psi)$; use a banded Jacobian sparsity pattern matrix (Section 2.4.1/Appendix A.4); and use the *numba* JIT compiler (Section 2.4.2/Appendix A.5). The water balance performance of this model, showing the cumulative change in storage against cumulative inflow (as infiltration at the surface) minus outflow (as drainage at the base) is plotted in Figure A1.





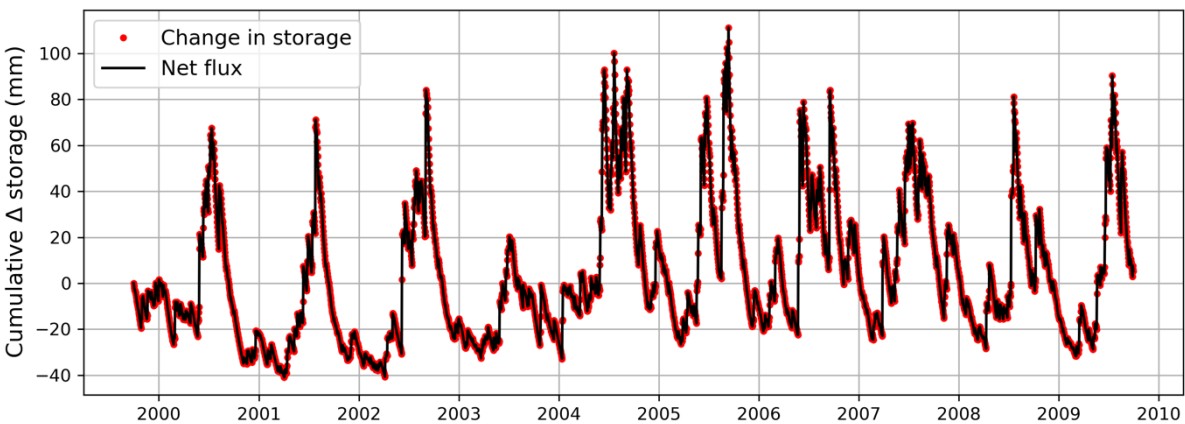

Figure A1. Water balance performance plot for 10-year infiltration experiment, with the best model configuration, showing
the cumulative change in storage in the profile, and the cumulative inflow minus outflow. The water balance bias was -0.018
mm, and the RMSE of the daily water balance errors was 8.06x10⁻⁵ mm.

*A.1 Alternative scipy ODE solvers*

Here we compare the alternative ODE solvers that were available in scipy at the time of writing, which includes *ode*, *odeint,*
and *solve_ivp*. These functions are alternative wrappers to classic ODE solvers written in Fortran, of which we consider here
VODE with the method BDF (Brown et al., 1989, available within "*ode*" and "*solve_ivp*"), and LSODA (available with all
three functions, Petzold, 1983). Note that for all solutions reported here we used the banded Jacobian sparsity pattern, with
the exception of *solve_ivp*-BDF, which only allows for the full Jacobian sparsity pattern to be defined and which we found
slowed the solution down – hence the results for the *solve_ivp*-BDF model do not use any information about the Jacobian
matrix.

| ODE solver | Method | runtime (s) | $\psi$ RMSE (m) | MB bias (mm) | MB RMSE (mm) |
|:---:|:---:|:---:|:---:|:---:|:---:|
| ode | BDF | 1.71 | 0.000* | -0.018 | 8.06E-05 |
| ode | LSODA | 1.96 | 0.000 | 0.001 | 7.11E-05 |
| odeint | LSODA | 2.69 | 0.000 | 0.000 | 7.04E-05 |
| solve_ivp | BDF | 7.45 | 0.001 | -0.594 | 3.33E-03 |
| solve_ivp | LSODA | 2.44 | 0.000 | 0.059 | 6.12E-04 |

Table A1. Model performance for the different ODE solvers/methods available in scipy.





We see that *solve_ivp* underperforms in accuracy, water balance, and efficiency. The *odeint* solver has the best performance
in terms of accuracy and water balance but is slower by a non-negligible amount. The *ode*-BDF method is the most efficient
but has slightly worse water balance performance – however, the water balance performance of all methods is extremely
good, and errors are negligible for practical purposes. We therefore chose *ode*-BDF as our preferred solution – but *ode*-
LSODA is also a good option. It is also possible to increase the error tolerances in the ODE solver, reduce the maximum
number of timesteps and increase the minimum timestep – all of which could result in a faster runtime at the cost of lower
accuracy/water balance closure.

### A.2 Alternative boundary flux calculation methods

As detailed in Section 2.3, there are alternative ways to calculate the boundary fluxes for use in the water balance
calculation. In Section 3.3, we developed a novel approach to calculating the boundary fluxes – the SFOM. In addition to
this method, we consider methods that calculate the boundary fluxes based on the output model states at either *reporting step*
or *calculation step* information. We also consider using forward, backward, or central difference approximations to integrate
the flux over a timestep (Eq. 18). The results of this analysis are provided in Table A2.

| Method | | runtime (s) | $\psi$ RMSE (m) | MB bias (mm) | MB RMSE (mm) |
|---|---|---|---|---|---|
| SFOM | | 1.71 | 0.000 | -0.018 | 8.06E-05 |
| Reporting step | central | 1.66 | 0.000 | 0.119 | 2.64E+00 |
| Reporting step | forward | 1.66 | 0.000 | 0.119 | 5.29E+00 |
| Reporting step | backward | 1.66 | 0.000 | 0.119 | 5.92E-02 |
| Calculation step | central | 5.24 | 0.000 | 0.046 | 1.13E-04 |
| Calculation step | forward | 5.24 | 0.000 | 0.046 | 3.24E-03 |
| Calculation step | backward | 5.24 | 0.000 | 0.046 | 3.14E-03 |

Table A2. Model performance using different calculation methods for the boundary fluxes.

Model state variables are unaffected by the different boundary flux calculation methods. The SFOM has the best water
balance performance, both in terms of bias and RMSE. Using calculation step level information results in good water
balance closure, with the central difference approximation giving the lowest errors. However, the efficiency of this is poor,





with runtimes increased by more than a factor of three. This is because many additional calculations need to be performed outside the ODE solver for each calculation time step. By default, the ODE solver allows up to 500 calculation time steps for every reporting timestep – so this is very inefficient. Calculating the boundary fluxes using reporting timestep information is very efficient, slightly faster than our method, but the water balance errors are significantly larger. These reporting step errors will increase with an increased reporting time step, as is shown in Figure 3. Overall then, the SFOM provides the

performance of using calculation step information without the loss of computational efficiency.

The key take home point here is that the easiest and most obvious approach to calculating the boundary fluxes is to use reporting step information. This is a bad idea – the mass balance errors are large, and if this is combined with other bad decisions (such as using discrete approximations for $C(\psi)$ as discussed in the next section), the results can be catastrophic

(water balance errors > 100 mm).

### A.3 Alternative estimation methods for $d\theta/d\psi$

When we use a parametric expression for $\theta(\psi)$, such as the van Genuchten equations (Eqs. 27-30), we can obtain an

analytical expression for $d\theta/d\psi$, as in Eq. 29, and this can be used to calculate $C(\psi)$ as implemented in RE in Eq. 5. However, depending on the numerical solution procedure that is adopted, this can lead to errors with mass conservation, and it is recommended by some researchers (Rathfelder & Abriola, 1994, Clark et al., 2021) that a discrete approximation is used for $d\theta/d\psi$, whereby

$$\frac{d\theta}{d\psi} = \frac{\theta_n - \theta_{n-1}}{\psi_n - \psi_{n-1}} \tag{31}$$


where here $n$ is a time index. This approach could be seen as equivalent to solving the mixed form of RE and can minimize water balance errors in the model that arise because the changes in $d\theta/d\psi$ over a time step are non-linear, as shown by Celia et al., (1990). However, it is necessary to apply this very carefully in the context of ATS methods. The values of $\theta_{n-1}$ and $\psi_{n-1}$ must be available from the previous calculation time step and not the previous reporting time step. If reporting timestep

information is used, the model will fail badly because as the calculation steps move forward in time over a reporting step, $C(\psi)$ is constantly referenced back to the beginning of the reporting step. This is clearly an erroneous approach, resulting in mass balance errors of > 100 mm for our problem. For the solver flux output method of calculating the boundary fluxes, it is necessary to output states at reporting steps, and therefore it is not possible to use the discrete approximation for $C(\psi)$. The results in Table WQ3 all use calculation step information. A more subtle issue is the order of the temporal integrator used by

the ODE solver, which can be specified by the user. Here, we use either 1st order or (variable) higher order (as determined by





the ODE solver) temporal integration methods. For the solver flux output method, we use higher order temporal integration. The results are given in Table A3.

| Method | | runtime (s) | $\psi$ RMSE (m) | MB bias (mm) | MB RMSE (mm) |
|---|---|---|---|---|---|
| Analytical $C(\psi)$ | Solver flux output method | 1.71 | 0.000 | -0.018 | 8.06E-05 |
| Analytical $C(\psi)$ | 1st Order | 36.25 | 0.000 | -0.49 | 4.97E-04 |
| Analytical $C(\psi)$ | High Order | 5.17 | 0.000 | 0.046 | 3.14E-03 |
| Discrete $C(\psi)$ | 1st Order | 44.62 | 0.000 | 0.317 | 1.92E-07 |
| Discrete $C(\psi)$ | High Order | 7.2 | 0.003 | 11.939 | 9.72E-03 |

Table A3. Model performance using different approaches to calculate $C(\psi)$.


We see in Table 3 that the discrete $C(\psi)$ approach works quite well for 1st order integration methods but is very slow. When higher order integration methods are used, the model is faster, but the mass balance is chronically degraded. We think that this happens because with higher order methods the model states evolve in a more complex manner (i.e., non-linear manner) over a calculation timestep, so the linear approximation in Eq. 28 is not good. It is noteworthy that the modelled $\psi$ values

were slightly modified using the discrete high order approach. For comparison purposes, we looked at using analytical representations of $C(\psi)$ with 1st order and higher order methods, and this time the higher order methods performed better. Overall, we recommend against using discrete $C(\psi)$ approximations, unless using a tailor-made ODE solver (such as Kavetski 2001, 2002a, 2002b).

*A.4 Alternative approaches to defining the Jacobian*

As described in Section 4.1, providing the ODE solver with information about the Jacobian matrix is reported to improve the solution efficiency. Here we compare three approaches: no information provided about the Jacobian; defining the Jacobian pattern; and defining the full Jacobian matrix. For the last case, this was complex to define for our method, and therefore this

was implemented for the high order reporting step solution procedure described in 4.3.2. The results are reported in Table A4.






| Method | | runtime (s) | $\psi$ RMSE (m) | MB bias (mm) | MB RMSE (mm) |
|---|---|---|---|---|---|
| Solver flux output method | Jacobian pattern | 1.71 | 0 | -0.018 | 8.06E-05 |
| | no Jacobian | 2.17 | 0 | -0.015 | 7.53E-05 |
| Reporting step flux calculation method | no Jacobian | 2.16 | 0 | 0.119 | 5.92E-02 |
| | Jacobian pattern | 1.63 | 0 | 0.119 | 5.92E-02 |
| | full Jacobian | 1.58 | 0 | 0.123 | 5.92E-02 |

Table A4. Model performance using different approaches to define the Jacobian matrix.

We see that for both model configurations, defining the banded Jacobian sparsity pattern matrix led to improvements in performance of around 20%. This is modest, but because it is trivial to define the banded matrix, this is worthwhile. For the reporting step model, when we defined the full Jacobian matrix, this led to a very slight improvement in performance over the banded solution (1.58 seconds vs 1.63 seconds). Defining the full Jacobian is challenging and requires an additional function/function call in the code – we therefore recommend against using the full Jacobian matrix and recommend instead defining the banded matrix.


### A.5 Running the model with and without numba JIT compilation

The best model configuration was also run with and without *numba* JIT compiler, and the result is shown in Table A5. It can be seen that *numba* has no impact on the model output (accuracy and mass balance are identical for each run) as 695 expected, but using *numba* improves the runtime by a factor of ~15. All other model runs reported in this paper use *numba*.

| Configuration | runtime (s) | $\psi$ RMSE (m) | MB bias (mm) | MB RMSE (mm) |
|---|---|---|---|---|
| With *numba* | 1.71 | 0.00 | -0.018 | 8.06E-05 |
| Without *numba* | 26.07 | 0.00 | -0.018 | 8.06E-05 |

Table A5. Model performance with and without *numba* JIT compliation.






**Code and/or data availability**

All of the scripts developed in this study are available from https://github.com/amireson/openRE, release v1.0.0, DOI: 10.5281/zenodo.6939855 (Ireson, 2022). The code is written in Python and MATLAB, and run using Makefiles, which reproduce Figures 2 – 7.


**Author contribution**

Ireson conceived of this study, wrote all the scripts (except the pseudospectral similarity solution, written in MATLAB by Mathias), performed the analysis and drafted the manuscript. Mathias came up with the idea for the proposed solver flux output method (SFOM). Clark, Spiteri and Mathias assisted with the study design and implementation and edited the

manuscript.

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
