# Peer review of "A simple, efficient, mass conservative approach to solving Richards' Equation (openRE, v1.0)"

_Geoscientific Model Development, 2022_

## Author Comment (AC1)

**Reviewer 1 comments**

Reviewer comments in black

Author's responses in blue

The paper entitled "A simple, efficient, mass conservative approach to solving Richards' Equation (openRE, v1.0)" outlines a straightforward implementation to solve the one-dimensional Richards' equation using off-the-shelf ODE solvers, with a novel amendment to effectively track the cumulative mass flux through the boundaries. The approach is rigorously compared to approaches and test cases from the literature. The paper is very well-written and structured. The contribution is somewhat novel (I have colleagues teaching solution of the advection dispersion equation using method of lines with basic ODE solvers at the undergraduate level; this is not a super-new idea), but the degree of rigour in assessment of the various libraries, tolerance and time step choices, and introduction of the SFOM flux tracking method puts this into the range of publishable contribution for a technical note in GMD.

Some minor nitpicking comments that the authors may want to consider here

1) the use of Q_j->j+1 (introduced in eqn 18) seems like subscript overkill - why not just Q_j ?

We prefer to be emphatic here about the meaning of this term. $Q_j$ is somewhat ambiguous, and while a precise definition can be given, it is much less likely that $Q_{j \to j+1}$ will be misunderstood.

2) it would be useful to report the domain extent and model simulation duration for Mathias' solution in section 3.1.3 (these are implicitly in the figure, but would provide a more complete problem statement in the text)

We including these details more precisely in Section 3.1.3, modifying the text to read:

This solution assumes a semi-infinite horizontal soil column ($0 \leq x < \infty$) with a uniform initial condition ($\psi(t = 0) = \psi_0$) and a type 1 boundary condition on the left boundary ($\psi(x = 0) = \psi_L$). The model was run for 100 minutes ($0 \leq t \leq 100$ min).

Some other things to consider in the future:

1) I envision the method of lines may perform even better in relation to other methods for cases with non-constant space steps and layering of different media. It would have

been nice to see a case study in this vein, but I would by no means require it here. Just something worth toying around with.

Thanks for this suggestion – we will surely look into this.

2) The use of arithmetic mean for calcualting hydraulic conductivity for the 1-D problem struck me as strange - the effective resistance to flow is typically treated using the harmonic mean for such problems by default (and this is well-documented even in the source they provided).

We agree that the harmonic mean is more typical, but either can be used, depending on assumptions about the structure of the porous medium. We will explore the harmonic mean in future work, but do not anticipate that this would cause any differences in the outcome of this paper.

3) It would be very interesting to see how this approach performs in the more relaxed domains simulated in land surface schemes, with inherently much larger space steps by default. That is, you are looking at the perfect limits against analytic solutions, but how does this approach do 'in the trenches' for practical problems where we can't afford the burden of 0.001s time steps and 0.0025 m space steps? Is it worth the effort of deploying for these types of problems?

This is a nice suggestion for exploring the utility of the numerical methods we have proposed in real world LSS, and we will explore this further – indeed similar methods are being implemented and tested by us in the SUMMA LSS.

I have been reviewing papers for 20 years and this is only the second initial submission where I have recommended acceptance 'as is'. Thanks for making my job as reviewer easy.

Thanks to Professor James Craig for this generous review. We greatly appreciate all the suggestions and feedback.

---

## Author Comment (AC2)

**Reviewer 2 comments**

Reviewer comments in black

Author's responses in blue

The manuscript focuses on numerical solutions to Richards equation, which is the de facto standard equation for simulating flows through variably saturated soils.

Richards equation has long attracted the attention of hydrogeologists/modellers with interest in numerical solutions, hence there is a rich history of publications on numerical (as well as analytical and semi-analytical) solutions for this equation. The integration of Richards equation in time using "time stepping" schemes is one of such common research topics.

Overall I enjoyed reading the manuscript and its topic fits well within the scope of GMD. It is great to see open source tools becoming more and more common for geophysical models. The implementations in Python are generally instructive (but see comments below).

Thanks for these positive comments above and critical comments that follow. The reviewer has raised a number of issues with terminology and clarity of presentation that we have addressed in our revised manuscript. We appreciate the rigorous review provided and the thoughtful comments. In the table below we provide individual responses to each of the reviewer's comments.

Unfortunately I have several major concerns with the current version of the manuscript:

1. The contribution of the study is stated in a confusing and potentially misleading way.

We understand and accept the reviewer's comments on this and have modified the text accordingly.

2. The benchmarking reported in the manuscript is carried out against common numerical schemes dating back to the 90s. However it is not acknowledged that many existing publications have already reported new (at the time) schemes that substantially outperform these common "old" schemes. Hence the claim of efficiency needs to be heavily qualified.

We again accept that we were not sufficiently clear about what aspects of our work is novel and what aspects are practical guidance for implementation. We have updated

the text, particularly the abstract and description of numerical methods, to address this.

3. The theory section is unfortunately not of high quality and makes numerous inaccurate / confusing statements. The description of at least one of the benchmarked methods is clearly incorrect (referred to as "no iteration" when the cited reference uses an iterative algorithm)

Again we understand this comment and have provided a detailed response to this below, and have clarified this in the revised manuscript.

These concerns are elaborated below (see "Detailed comments").

I see the real contributions of this work as follows:

- the new method to quantify the boundary fluxes that reduces water balance errors without negative impacts on model runtimes. I like the idea of adding an extra mass balance term to the vector of unknowns for the ODE solver to keep a more accurate track of the mass balance fluxes. This is a neat idea for off-the-shelf methjds.

- an open source implementation of common algorithms for (1D) Richards equation. This is indeed very nice for learning how these algorithms operate. However here please double-check/correct your implementation of the pressure-form solution against the description in the cited Celia et al 1990 publication

- benchmarking of these common algorithms against selected alternatives. This contribution is also useful but should be more carefully delineated. First the benchmarking using CPU time depends heavily on how much care is taken to optimise the code. Second the methods included in the comparison are indeed representative of common methods for Richards equation, but many methods (reported in the literature) that outperform these common methods are not included in the comparison. This is a pretty pertinent piece of information to be noted to avoid confusing or even misleading readers.

We agree with the reviewer's summary of what our contributions are, and we have attempted to better clarify this in the revised manuscript.

I recommend a major revision to address these concerns, and rebalance the manuscript towards the new findings.

Detailed review comments

1. Contribution

The results presented upfront as the major findings are already well known and established in the literature on Richards equation.

For example, consider the first sentence of the abstract, which states "We show that a simple numerical solution procedure – namely the method of lines combined with an off-the-shelf ODE solver – can provide efficient, mass conservative solutions to the pressure-head form of Richards' equation".

In the Richards eqn literature this result was established as far back as 1997 by Tocci et al who indeed used the method of lines, an off-the-shelf ODE solver (DASPK) - and applied it to the pressure form to achieve good mass balance and high numerical accuracy.

From the Abstract of Tocci et al's paper: "We show how a differential algebraic equation implementation of the method of lines can give solutions to [NB: pressure-form] RE that are accurate, have good mass balance properties, explicitly control temporal truncation error, and are more economical than standard approaches for a wide range of solution accuracies".

There is also the follow-up paper by Miller et al (1998) which further developed these results. The studies by Tocci et al and Miller et al are well cited and amongst the classics for the time integration of Richards equation. So this is by no means some poorly recognised result being brought to light in the new manuscript.

It would be important to clarify how the submitted manuscript contributes beyond this existing knowledge.

We agree with the reviewer that presenting a method of lines solution for the pressure form of Richards Equation is not a novel contribution – we phrased the first sentence poorly. We are aware that this has been done by Tocci et al and Miller et al. We have modified the first sentence to read

"A simple numerical solution procedure – namely the method of lines combined with an off-the-shelf ODE solver – was shown in previous work to provide efficient, mass conservative solutions to the pressure-head form of Richards' equation. We implement such a solution in our model openRE. We developed a novel method to quantify the boundary fluxes that reduce water balance errors without negative impacts on model runtimes – the *Solver Flux Output Method* (SFOM)."

Please note – we do not include references in the abstract, but we have referenced all suggested work in the body of the paper. Through the text we try to make it clear that we are "implementing" an existing method (albeit with different off the shelf solvers than have been previously reported), and aside from the mass balance calculation, this is a practical contribution, not a theoretical one.

The superior mass balance of the Modified Picard method (applied to the mixed form of Richards equation), which is stated as another key finding in the Abstract (line 17), is also not a new result. The Modified Picard method is by construction mass conservative, as designed by Celia et al in 1990 (and further explained by Rathfelder and Abriola 1994). This result is also common knowledge in the field and stated in the abstract of the 1990 paper which has 1000+ citations.

We are not saying that it is a new insight that the Modified Picard Iteration method has good water balance performance (we agree with the reviewer's comments on this point). We state that this method has superior water balance performance to the alternative methods we tried – that may be unsurprising, but it is still a fair and correct statement. There doesn't seem to us to be anything unreasonable about the language we have used in this part of the abstract.

Line 38. The first objective is stated as "present a simple and practical approach to solve RE that is efficient, mass conservative," - please clarify if this is an existing approach or a new approach (and if so how it differs from existing approaches). See comments above on the use of existing off-the-shelf solvers for Richards eqn.

We have changed the word "present" to "implement" which softens any potential implication (which we do not ourselves perceive) that we are claiming this is a completely novel method.

Abstract Line 18-19 "Our solution has comparable run-times to Hydrus and better water balance performance (though both models have excellent water balance closure for all the problems we considered)."

Please clarify what is meant by "our solution". Do you mean an off-the-shelf (i.e. pre-existing) ODE algorithm applied to Richard eqn? If so please detail which off-the-shelf algorithm is being used.

"Our solution" is openRE, and is described in detail in this paper, and made available openly through the "openRE" repository that we provide. We clarify this with the text in the abstract. The details of our solution includes the form of the PDE we solve, the python-scipy ODE solver we use, the way we implement the Jacobian pattern, the way to calculate the water balance, and the just-in-time compiler that we use. All of

these "subtle decisions" are described and evaluated in the paper (including in Appendix A).

Please also clarify the term "better mass balance performance". Given that one of the contributions of the study is a new approach for estimating mass balance, does "mass balance performance" refer to actual mass balance errors, or to how are they estimated within the algorithm? Also please clarify if the same approach for estimating mass balance errors is used in the benchmarking of the algorithms - as that seems pertinent for benchmarking.

The details are given in the paper. These are actual mass balance errors assessed on the basis of the model outputs (for all models) after the model run is completed. We don't see any need for modification of this sentence.

Line 97 "though it has been shown that mass conservative solutions are possible (Rathfelder and Abriola, 1994, Tocci et al., 1997). This issue will be explored in detail in this paper."

Given that this is one of the aims, please make it clearer across the paper, as well as in the Abstract/Conclusions, how this study extends what is already known on this topic from earlier work.

Our findings corroborate those of others that have shown $\psi$ forms of RE can be mass conservative. This is not a novel contribution, nor do we claim it is. In our appendix A2 and A3 we show how poorly implemented $\psi$ forms of RE are not mass conservative. This maybe a useful practical advice, though again we are not claiming novelty with this. To make the text in question less ambiguous we replaced

"This issue will be explored in detail in this paper."

With

"We carefully assess the mass balance performance of our model in this paper."

Section 2.3 Mass balance closure

This section presents one of the interesting/novel contributions of the study, which is the embedding of the mass balance calculation into the off-the-shelf ODE algorithm.

However this contribution is not so well delineated within this section. For clarity it would make sense to have this contribution in its own (sub)section, rather than

presented alongside existing approaches. Then it would be clearer to compare the new approaches to the existing approaches.

In many ways this is the centerpiece of the work (and its contribution as mentioned in the abstract and objectives) - so worth making it stand out.

It is also not clear from the current presentation (at least the Abstract/Conclusions) what are the practical improvements in mass balance accuracy. Can you quantify these improvements in the key sections of the manuscript? This would help re-orient the presentation towards the new results.

Thanks for this comment - we agree that this is an important and novel component of this study. We term this method the Solver Flux Output Method (SFOM) and this was very prominently described in the conclusions. It was not, however, prominent in the abstract, which we have now updated to address this – please see the first three sentences of the revised abstract.

2. The algorithms included in the benchmarking are representative of current practices (as in "practical work" rather than "research work"), but do not include algorithms found to be the most efficient in many of the studies cited in the manuscript.

The main method proposed in this work is an off-the-shelf DE solver applied to the pressure form of Richards equation. This solver uses truncation error control through adaptive time stepping.

The benchmarking is done against fixed-step implementations of methods dating back to 1990s, namely the Standard Picard (likely/possibly implemented inconsistently with the cited original paper - see later comments) and Modified Picard (which is arguably the de facto standard in practical modelling tools).

Note that the manuscript presents Hydrus 1D as a distinct method, however (from memory) this software also implements the Modified Picard iteration. Indeed, from the Hydrus 1D description in https://www.ars.usda.gov/pacific-west-area/riverside-ca/agricultural-water-efficiency-and-salinity-research-unit/docs/model/hydrus-1d-model/, "The water content term is evaluated using the mass conservative method [NB: Modified Picard] proposed by Celia et al. [1990]."

So it is hardly surprising that the modern off-the-shelf method is more efficient.

The same comparisons of adaptive methods have been carried out in 1997 by Tocci et al (and their subsequent work using high-order DASPK solvers), by Kavetski et al 2001 (and subsequent work using first/second order implicit methods), by Baron et al (2017), and others. In all these publications adaptive methods were found to outperform fixed-step implementations (and heuristic time stepping implementations). And at least some of these publications have already compared multiple adaptive time stepping methods.

So when the manuscript claims the proposed method is efficient, it should be noted that this is in comparison to methods that have already been found to be inefficient decades ago in the research literature. This seems a rather important point to make to avoid confusing or potentially misleading the readers.

We do agree with the reviewer that the algorithm we implement to solve our model is a practical approach, rather than novel contribution. We have softened the text in various places to try to be clear about this – notably the abstract.

We note that it was not obvious to us, going into this work, that using the python interpreted language and scipy ODE solvers would allow us to achieve runtimes equivalent to Hydrus – a well-established and widely used model that has been improved over decades and uses a compiled language. We were pleasantly surprised by this.

3. Presentation of background theory / methods

The classification of methods on page 4 (lines 105-132) is quite confused and there are many substantial inaccuracies in the descriptions.

Adaptive time stepping (ATS) methods are not a different class of methods to "iterative methods that take one step" or to "one-step methods without iteration".

The name "methods that take one step" is non-standard and quite confusing because all the methods here take multiple steps (indeed thousands to millions of them).

I appreciate that classifying the wide range of numerical methods into neat self-contained categories is not always easy, but there are plenty of good textbook-level references that provide more rigorous classifications and standard naming conventions. For example the standard term is "single-step methods" in reference to methods that carry out a time step using information from that time step alone (in contrast to "multi-step" methods that carry information from preceding time steps).

We mainly accept these criticisms and appreciate the feedback which has allowed to to (hopefully) present this more rigorous and clear language. The main problem was the term "one-step", which we agree is the wrong term, and we agree that "fixed-step" is an improvement.

Another issue that was not clear was our claim that Celia et al. (1990) present a non-iterative solution – we have addressed that in the response below (in short – even though Celia et al did not do this, their Eqn 4 can be solved directly, which is the non-iterative approach that we implement – and we include this purely for educational purposes).

We made extensive edits to Section 2 after Line 105 to address these comments.

Line 107 makes a very confusing statement that "methods with no iteration" are defined as methods "where a single evaluation of Eq. (9) is performed for each time step".

Eq 9 is the continuous time ODE - how does one evaluate it once for each time step? If you are referring to the RHS of that equation - that is also not correct because many methods without iterations evaluate the RHS several times within each time step.

Fair comment – this text was removed.

Line 111: "Semi-implicit one-step methods may employ the Thomas Algorithm, or similar, to solve the tri-diagonal matrix problem".

This statement is problematic for many reasons, e.g. it applies only to 1D problems with a particular spatial discretisation (which yields tridiagonal matrices). Further, iterative methods for such 1D problems will also typically use the Thomas algorithm within their iterations, otherwise they would be a rather inefficient implementation!

We agree with the reviewer here – we have deleted this part of the sentence, and generally improved this paragraph.

Line 113. "which is problematic due to the non-linear dependence of K and C on psi (Celia et al., 1990, reproduced below)".

This statement seems to confuse explicit methods (which indeed may not be a good practical choice for Richards eqn) and semi-implicit methods (which include some of the most efficient methods reported for Richards equation).

We agree this was poorly written – we have modified this paragraph to make it clearer. To be clear, it is non-iterative methods that have this generic limitation, since they do not attempt make the solution converge that have this limitation.

Line 115: "The iterations allow the solution to find the correct values of K and C at the next time step"

Strange to refer to any of these values as "correct" - they are all approximations.

Agreed – we changed "the correct" to "more representative"

Line 128: What is a "black-box" solver in this context?

This is Kavetski's term (as the citation and quotations make clear), and we take it to mean an off the shelf solver which may be applied by the user without understanding the details of how it works internally. We think the term should be readily understandable.

Overall the text/classification on 105-133 needs to be more or less rewritten from scratch. Here I would recommend contrasting fixed-step vs adaptive-step methods, and then contrasting the methods used within the time steps, etc.

We have tried to address all of these comments with edits to the text and we have adopted the term "fixed-step". We now consider NIM (Fixed-step Non-iterative methods), FIM (Fixed-step Iterative Methods) and ATS (Adaptive Time Stepping) methods.

Description of methods used in benchmarking

Line 149-151. "typical one-step method (first-order backward Euler implicit solution), and an excellent iterative method in their improved Modified Picard Iteration Method (MPM) solution"

This description is very confusing and stems from the earlier confusion on lines 105-133. Both the backward Euler method ("Standard Picard") and the Modified Picard method used by Celia et al (1990) are single-step *iterative* methods. It is strange - and very confusing! - to refer to one as "one-step" and to the other as "iterative".

We think our edits to this text will make this clearer. Note in particular our response to the next comment below.

Line 367. "the 'No iteration scheme' (Note: referenced earlier to Celia et al 1990) uses the psi form of RE and solves the problem with a single backward implicit step and no iteration (which we achieved using the Thomas Algorithm)."

As noted earlier, all schemes used by Celia et al 1990 are iterative - for the Standard Picard method (backward Euler solution of pressure form in Celia et al study), this should be clear from Celia et al's equations 4-5. So this sentence describing one of the schemes being benchmarked is incorrect in a very obvious way.

Further, as noted earlier the Thomas Algorithm is also used in the Modified Picard iteration for 1D problems - or at least *should be used*, as otherwise there would almost certainly be a major loss of efficiency. See equation 17 in Celia et al where the tridiagonal coupling between the unknowns can be seen.

The reviewer is correct that Celia presents results from two iterative solutions (Celia's Fig 1 shows the Picard Iteration (PI) solution and their Fig 5 shows the Modified Picard Iteration (MPM) solution, both of which are reproduced in our Figure 2 using our implementation of Celia's methods).

However, Celia's Equation 4 can also be solved directly, and this is what we describe as the NIM (i.e. non-iterative) solution. We include this for teaching purposes – it is useful for students to understand and see the limitations of non-iterative methods. We also show these results in our Figure 2. It is also worth pointing out that some existing land surface models actually use non-iterative solutions in their implementation of RE – so some better education around this problem might be useful.

Line 158 "All solutions make use of the Thomas Algorithm to solve the tridiagonal linear system arising from the implicit method."

Tridiagonal systems arise only in 1D (in space) problems with certain simple spatial discretisations (e.g. second order central finite differences). To avoid confusion, earlier text should make it clearer you are restricting your attention to 1D problems and the specific spatial discretisation.

Specifically, Line 59 and the rest of the presentation would be clearer if you stated "*1D* vertical flow", and then somewhere around Line 100 would be important to state that you are using second-order differencing in space, which yields tridiagonal matrices.

This is a fair comment – we should have been clearer that we focus on 1D RE. We adopted the reviewer's first suggestion here (1D flow). We don't feel it is necessary or helpful to state at this point in the text that we are using second-order differencing in space or that this yields a tridiagonal matrix.

Line 394. "This is perhaps an under-appreciated limitation of Celia's MPM solution, and solutions to the mixed form of RE generally – which is that mass balance is a necessary but insufficient criterion for model performance assessment, and truncation errors can still be present in the fluxes with perfect water balance closure"

Please clarify briefly why you consider this to be under-appreciated?

The issue of errors in the simulated fluxes associated with Celia's MPM solution is not addressed in Celia's paper, where one can get the impression that the MPM solutions are good irrespective of the time step. Our analysis in Figure 3 shows that this is not the case. We have not seen this shown elsewhere in the literature. Perhaps it is obvious to the reviewer, but we think other readers will benefit from this simple demonstration. Our language is also very moderate here – "… this is perhaps an under-appreciated limitation…" – this seems quite reasonable to us.

Section 2.4 "Improving efficiency" / subsection 2.4.1 "Defining the Jacobian pattern"

I think this section would make good sense when working with 2D/3D Richards equation where the (Jacobian) matrices are large and sparse. However as far as I understand (and note above) the authors are working with 1D Richards equation and second order spatial finite differences. If so, then the matrices appearing in the discretised (Picard iteration) equations are all tridiagonal and the Jacobian matrix is also tridiagonal.

For tridiagonal matrices, the Thomas algorithm is standard and will be more efficient than alternatives. So I don't get the point of experimenting with different Jacobian patterns given that the actual Jacobian pattern is known exactly and the best linear solver algorithm for it is also well known.

In this section 2.4 we are describing the benefits of providing the off-the-shelf ODE solver with information about the pattern of the Jacobian matrix. Doing this is extremely simple and has significant effects on runtimes – this is the simple point we are making here. We are not talking here about implementations of the Thomas Algorithm or any other algorithm that the user must implement themselves. Again – a practical and not a theoretical contribution.

Other

The comparison of compiled languages (listed here as Fortran/C) vs interpreted languages (Python) in Section 2.4.2 is rather subjective and should be indicated as such. For example, line 332 essentially states that interpreted languages are "easier to learn, with excellent teaching resources widely and freely available".

The first of these statements (ease of learning) is clearly subjective - the authors should qualify this as either being their opinion, or (preferably) provide references to some kind of surveys on this topic. The latter two claims are puzzling because there are certainly plenty of excellent resources for Fortran and C, and free compilers are available for both languages. Indeed, the only non-freely available language that comes to mind is Matlab, which happens to be an interpreted language.

Worth mentioning C++ which is perhaps the most modern and widely used compiled language, and for which once again excellent resources and free compilers are commonplace.

This comment is not to deny the obvious appeal of intepreted languages, but a manuscript that focuses on computational software can surely use more accurate and defensible statements when discussing language choices.

We are happy to state that this is our opinion. We modified the text to read:

"they are, at least in our opinion, easier to learn, with excellent teaching resources widely and freely available"

We also listed C++ as an example of a compiled language.

Lines 338-344: A citation to a numba reference would seem appropriate here.

Agreed – we have added the following citation:

"Lam, S. K., Pitrou, A., & Seibert, S. Numba: A LLVM-based Python JIT compiler. Proceedings of the Second Workshop on the LLVM Compiler Infrastructure in HPC, 1–6. https://doi.org/10.1145/2833157.2833162, 2015"

---

## Author Response (AR2)

**Topical editor comments**

Reviewer comments in black

Author's responses in blue

Dear Authors,

Thank you for having carefully addressed the comments from the two reviewers. The modified version of the manuscript is in good shape and could certainly be accepted for publication in GMD upon addressing the minor revision suggestions I list hereafter.

Best regards,

Ludovic Räss - Topical Editor

Thanks for the useful suggestions. We have adopted all suggested edits, and, in particular, have removed some of our potentially controversial comments about interpreted vs compiled languages. We will look into Julia in future work. Thanks again, Ludovic, for your kind support dealing with our paper.

Specific comments:

l.20: Why only listing Matlab and Python as interpreted languages? Did you try these two? If no, I would suggest to only keep interpreted languages without example.

Yes – we have coded up solutions to certain problems discussed in the paper in both Python and MATLAB. We have updated the model repo associated with this paper to include a MATLAB version of the model (https://github.com/amireson/openRE)

Also, as a general comment, SciPy is mostly compiled as most of SciPy's functions actually call compiled C or Fortran code

We agree.

l.23: "code is short and simple" -> concise?

Agreed – change implemented.

l.141-147: Did you look into and/or try Julia as language? Julia has an interesting DifferentialEquation.jl environment that exactly addresses solving PDEs using the method of lines. Could be worth checking out briefly and also mention it in here.

We will investigate this, but we have not had an opportunity to try this out yet. We cannot mention this in the current manuscript, without further exploration.

Fig.1: Are you using a staggered grid? Of so, maybe worth mentioning it referencing it. Staggering will actually make the scheme conservative.

We had originally used the term "block-centered grid" which is common, but we have now updated this to "cell-centered grid" following Bear & Cheng 2010, where this type of grid is clearly described. This is precisely equivalent to a staggered grid, but that terminology is not, to our knowledge, used in our field. We updated our terminology from "block" to "cell" throughout, and we added the note on line 170: "(note, this is equivalent to what is called a "staggered grid" in fluid dynamics)".

Section 2.4.2: The statements reported here are somewhat inaccurate:

- "Interpreted languages are compiled on the fly, meaning every individual line of code is compiled at run-time" -> not true. Interpreted languages are interpreted, not compiled. Some functions maybe be written in C, Fortran and (pre-)compiled (e.g. vectorised operations in Matlab, packages in Python such as Numba) leading to better performance.

- "By contrast, compiled languages are more efficient because the compilation and running of the code are separated, and you typically only have to compile the code once before it can be run many times" -> not true. Compiled languages are more efficient because they are compiled. Meaning, the compiler can specialise the expression, make lots of optimisations (loop fusion, and much more) as no interpretation needs to happen any more upon compilation.

Responding here to these two points - it is debatable whether it is fair to characterise an interpreted language as compiling each line at runtime – but we do not wish to say anything controversial in our paper that will detract from the message. We also agree with your second point about compilation. Therefore, we accept the criticism, and we have reduced and simplified the text in question to read: "Interpreted languages are not pre-compiled and are hence slower to execute than compiled languages. A nice compromise between the simplicity of interpreted languages and efficiency of compiled languages is to use a just-in-time compiler."

- "A nice compromise between the simplicity of interpreted languages and efficiency of compiled languages is to use a just-in-time compiler. In Python, the numba library is such a just-in-time compiler (Lam et al., 2015)." -> That's correct.

Glad we agree on this.

- "Numba compiles all the lines of the Python code once at the start of the runtime, and then all subsequent calls to the code run much faster." -> Numba compiles only the subset of the code or functions marked as such.

Agreed – we modified the text to read: "Numba compiles selected Python functions once at the start of the runtime, and then all subsequent calls to the code run much faster."

As FYI, Julia would compile the entire code. Maybe worth mentioning it and even checking it out.

Thank you – we will certainly do this in future.

Section 3: for criteria iv) and v), it would be good to report which hardware you were running on (CPU, single/multi-cores, RAM, etc...) and elaborate a little bit more on the concept of "simplicity" of the code.

To address this comment in the introductory paragraph of section 3 (line 351) we added "For the purposes of comparing efficiency ($iv$), all simulations were run on the same laptop computer and we report the runtimes as a measure of relative performance. For the purposes of comparing simplicity of the code ($v$) we use a very simple metric of lines of code, which is reported above in Section 2."

Fig.6: May be good to annotate separate sub-panels, and fix the overlapping y-thick labels for the lower subplots

We modified the units in the lower plot to correct the overlapping y-ticks. We added annotations to identify each subplot and refer to these in the revised caption. We also added annotations to Fig 7.

---

## Author Response (AR3)

**Author's response**

There were no further edits requested. In the current version of the manuscript the only edit made was to corrected the DOI number of the model repository, which is version 1.0.1.

Andrew Ireson